# Capture-C reveals preformed chromatin interactions between HIF-binding sites and distant promoters

James L Platt[1], Rafik Salama[1], James Smythies[1], Hani Choudhry[2], James OJ Davies[3], Jim R Hughes[3], Peter J Ratcliffe[4,**] & David R Mole[1,*]

## Abstract

Hypoxia-inducible factor (HIF) directs an extensive transcriptional cascade that transduces numerous adaptive responses to hypoxia. Pan-genomic analyses, using chromatin immunoprecipitation and transcript profiling, have revealed large numbers of HIF-binding sites that are generally associated with hypoxia-inducible transcripts, even over long chromosomal distances. However, these studies do not define the specific targets of HIF-binding sites and do not reveal how induction of HIF affects chromatin conformation over distantly connected functional elements. To address these questions, we deployed a recently developed chromosome conformation assay that enables simultaneous high-resolution analyses from multiple viewpoints. These assays defined specific long-range interactions between intergenic HIF-binding regions and one or more promoters of hypoxia-inducible genes, revealing the existence of multiple enhancer–promoter, promoter–enhancer, and enhancer–enhancer interactions. However, neither short-term activation of HIF by hypoxia, nor long-term stabilization of HIF in von Hippel–Lindau (VHL)-defective cells greatly alters these interactions, indicating that at least under these conditions, HIF can operate on preexisting patterns of chromatin–chromatin interactions that define potential transcriptional targets and permit rapid gene activation by hypoxic stress.

**Keywords** Capture-C; chromatin; *cis*-interaction; HIF; Hypoxia
**Subject Categories** Chromatin, Epigenetics, Genomics & Functional Genomics; Transcription

## Introduction

Hypoxia-inducible factor (HIF) is the key transcription factor regulating transcriptional responses to hypoxia and is central to the pathogenesis of many human diseases including cancer. The DNA binding complex compromises an oxygen-regulated alpha-subunit, which dimerizes with a constitutively expressed beta-subunit and binds to *cis*-acting hypoxia response elements containing a core "RCGTG" consensus sequence [1,2]. Activation of HIF directs a broad range of inducible responses, which vary according to cellular and pathophysiological context [3], raising questions as to how different responses are specified.

Recent advances in high-throughput DNA sequencing have facilitated pan-genomic analyses of the architecture of HIF transcriptional response and are beginning to shed light on these processes [4–7]. These studies have revealed hundreds to thousands of HIF-binding sites across the genome. As expected, a proportion of these sites clusters with annotated promoters. However, approximately 30% of HIF-1 & 50% of HIF-2 binding sites are more than 10 kb away from their nearest annotated gene [7]. Tests of association with hypoxia-inducible transcripts, such as gene set enrichment analyses (GSEA), have demonstrated clear statistical association with positive effects on gene expression, suggesting that at least some of these loci are hypoxia-inducible transcriptional enhancers [7]. However, with the exception of a small number of loci [8,9], physical association with a hypoxia-inducible promoter has not been defined. Nor is it known whether any such associations are induced following exposure of cells to hypoxia and/or the binding of HIF to DNA or whether they are present prior to the induction of HIF and contribute to the cell-type-specific operation of the HIF transcriptional response. These questions are of interest both mechanistically, since hypoxia has been reported to affect chromatin structure [10–13], and medically, since specific therapeutic modulation (both positive and negative) of components of the hypoxic transcriptional response is a major focus for pharmaceutical development [14,15].

1   Henry Wellcome Building for Molecular Physiology, University of Oxford, Oxford, UK
2   Department of Biochemistry, Faculty of Science, Center of Innovation in Personalized Medicine, King Fahd Center for Medical Research, King Abdulaziz University, Jeddah, Saudi Arabia
3   Medical Research Council Molecular Haematology Unit, Weatherall Institute of Molecular Medicine, University of Oxford, Oxford, UK
4   Target Discovery Institute, University of Oxford, Oxford, UK
    *Corresponding author. Tel: +44 1865 287788; E-mail: drmole@well.ox.ac.uk
    **Corresponding author. Tel: +44 1865 612680; E-mail: Peter.Ratcliffe@ndm.ox.ac.uk

To address these, and related questions as to the number and range of interactions made by HIF-binding sequences under different inducing conditions, we deployed a recently developed multiplexed assay of chromatin conformation [16,17] in normoxic and hypoxic cells, and in renal cancer cells in which HIF is activated in normoxia by inactivation of the von Hippel–Lindau (pVHL) E3 ubiquitin ligase [18]. This technology enabled the detection of an average of approximately 6,000 unique ligation junctions at each of 38 viewpoints overlapping selected promoter-distant HIF-binding sites and gene promoters. Our findings revealed that these intergenic HIF-binding regions commonly interact with the promoters of hypoxia-inducible genes. They demonstrated *cis*-acting chromatin interactions from HIF-binding regions to multiple types of DNA element, including both the promoters of hypoxia-inducible genes and other distant elements over distances of up to 250 kb. The findings also revealed that most intergenic regions that bind HIF in hypoxic cells bear the marks of active enhancers in normoxic cells and that *cis*-interactions between these sites and hypoxia responsive promoters are already established, with few changes occurring during induction of the HIF response by hypoxia. Taken together with other recent analyses of the HIF response, the results support a model of gene regulation whereby many HIF-binding sites and their target promoters are accessible, functionally annotated, active, and associated in close proximity prior to HIF binding, ready to effect a rapid transcriptional response through release of promoter-paused RNApol2.

## Results and Discussion

### Chromatin environment at distal HIF-binding sites

To investigate HIF-binding sites that are distant from annotated gene promoters or transcribed regions of the genome [7], we first defined whether these distal sites carry histone modifications consistent with enhancers rather than unidentified promoters. ChIP-seq was performed in MCF-7 cells, incubated in normoxia or hypoxia (0.5% oxygen, 16 h), using antibodies to H3K4me1 and H3K27ac, and analyzed together with RNA-seq and H3K4me3, RNApol2 and HIF ChIP-seq datasets obtained previously under similar conditions [4] as well as DNase-seq analyses published by ENCODE (GSE32970) (Fig 1A–N). Importantly, HIF binding was highly inducible by hypoxia, with almost no detectable levels of normoxic HIF-1β binding (Fig 1A and B). HIF-binding sites that were more than 10 kb from annotated transcriptional start sites were associated with negligible levels of RNA expression compared with gene bodies (Fig 1C and D). We were unable to confidently observe eRNAs, most probably because the total RNA-seq data were not sequenced to sufficient depth. These promoter-distal HIF-binding sites were highly accessible compared to promoters and were strongly enriched for the enhancer mark H3K4me1, with lower levels of H3K4me3 [19,20] (Fig 1E–J).

Although levels of H3K4me1 were unchanged at promoter-distal HIF-binding sites in hypoxia (Fig 1G), H3K27ac and RNApol2 signals were increased (Fig 1K and M) consistent with increased enhancer activity in hypoxia [21]. However, H3K27ac and RNApol2 were also present at these sites in normoxia before HIF binds. Similarly, RNApol2 was present across the gene bodies of hypoxia-regulated genes (Fig 1N) indicating that they are transcribed, at lower levels

(Fig 1D), before HIF is stabilized in hypoxia. This supports a model in which promoter-distant HIF binding can occur at enhancers that are already partially active in normoxic cells.

### Chromatin interactions with HIF-bound enhancers

To determine how these putative HIF-bound enhancers act on their target promoters, we next examined chromatin *cis*-interactions with a subset of promoter-distant, HIF-binding sites (Appendix Table S1). As expected, these co-localized with ChIP-seq signals for H3K4me1 and H3K27ac (e.g., Fig 2A). We first studied hypoxic MCF-7 cells and analyzed *cis*-interactions with 18 intergenic HIF-binding sites that were > 10 kb (median 74 kb, mean 125 kb) away from any annotated promoter, using Capture-C [16,17]. Capture-C is a derivative of the chromatin conformation capture (3C) technique [22], coupled with oligonucleotide enrichment and high-throughput sequencing. Capture-C allows discovery of previously unknown distant interacting elements (promoters, enhancers, CTCF sites, etc.) from multiple "viewpoints" (in this case HIF-binding sites) in parallel and at high resolution. Each experiment was performed in duplicate and demonstrated a high degree of reproducibility (Appendix Fig S1).

At 14 of the 18 sites, strong enrichment of sequences at the "viewpoint" site was observed confirming efficient capture; sites with poor capture efficiencies were excluded from further analysis. Regions that had a re-ligation frequency that was significantly greater than a background, modeled on the average distance-dependent signal decay, were defined as sites of significant *cis*-interaction. Each HIF-binding site showed significant interaction with an average of seven sites (Fig 2B). These *cis*-interacting sites were then correlated with UCSC gene annotations and ChIP-seq analyses of H3K4me3, H3K4me1, H3K27ac, RNApol2, and CTCF binding to define functionally annotated regions. Approximately 50% of *cis*-interacting sites contained one or more of the above marks (Fig 2B). *Cis*-interacting elements lay predominantly within 250 kb of the HIF-binding site (Fig 2C) and all were located within the same topologically associating domain (TAD) [23].

*Cis*-interactions were identified between HIF-bound enhancers and at least one annotated gene promoter lying up to 150 kb from the HIF-binding site at nine of the 14 HIF-binding sites assayed. Notably, this was not always the nearest gene. At three sites, *cis*-interaction with the promoter of a hypoxia-regulated gene was observed, although the number of interactions was below the level required to be statistically significant. At only two out of 14 HIF-binding sites was no association with the promoter of a gene detected. Both had evidence for strong HIF binding, although the closest hypoxia-regulated gene was > 1.5 Mb away. Whether these two sites represent a distinct functional class of HIF-binding sites will require further analysis. Three sites where *cis*-interactions with more than one promoter were seen were all associated with a co-regulated coding and noncoding gene pairing (e.g., Fig 2A).

Genes whose promoters interacted with distant HIF-binding sites were enriched among hypoxia-upregulated transcripts in RNA-seq analyses as determined by Gene Set Enrichment Analysis (GSEA) [24], using a ranking metric combining fold-change in hypoxia and significance [25] (Fig 2D). By comparison, adjacent genes whose promoters did not interact were not enriched for hypoxic regulation (Fig 2E) despite the presence of H3K4me3, RNApol2, and RNA-seq

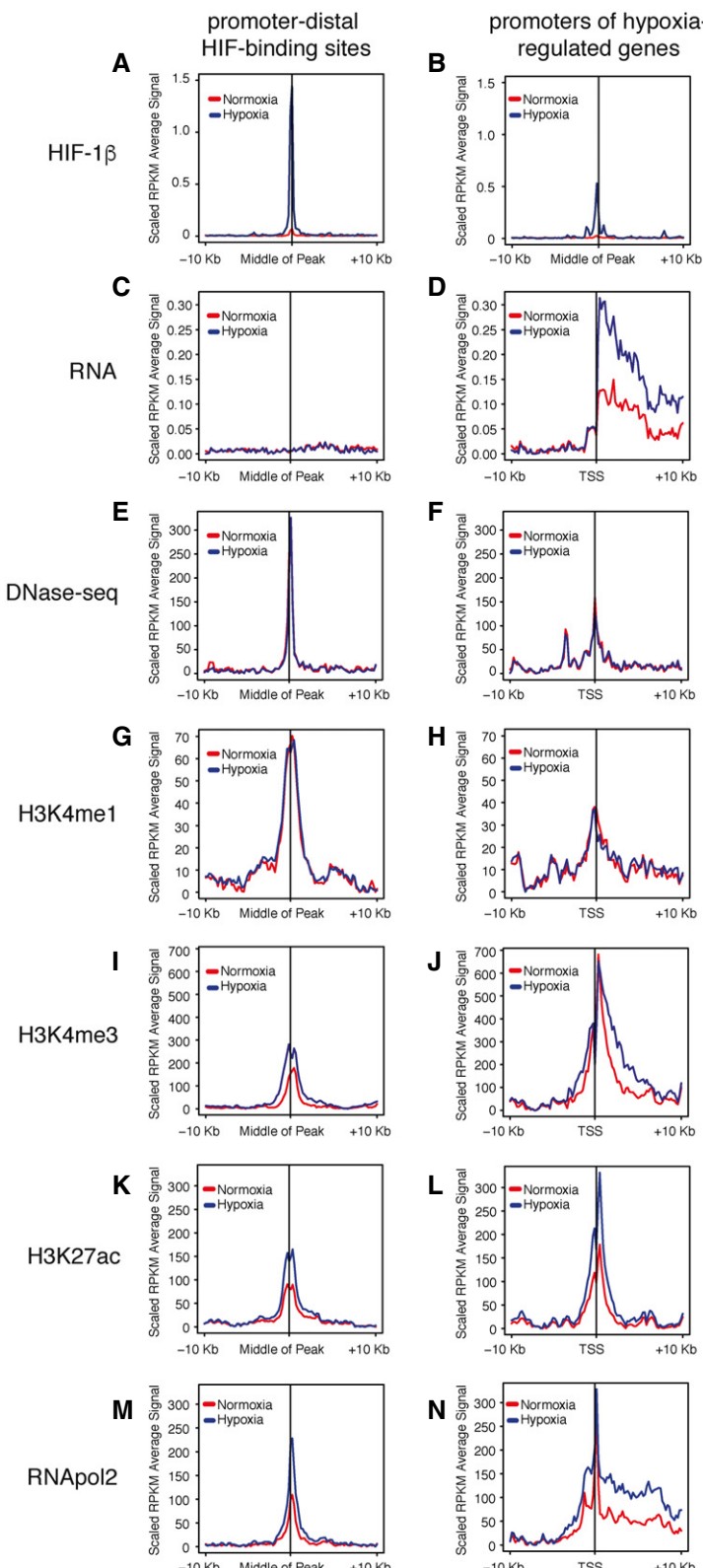

**Figure 1.   HIF-bound enhancers are in a poised state in normoxia.**

A–N    Averaged signals at promoter-distal (> 10 kb) HIF-binding sites and at the promoters of hypoxia-regulated genes for comparison are shown for HIF-1β ChIP-seq (A, B), total RNA-seq (C, D), DNase-seq (E, F), H3K4me1 ChIP-seq (G, H), H3K4me3 ChIP-seq (I, J), H3K27ac ChIP-seq (K, L), and RNApol2 ChIP-seq (M, N). Median RPKM (reads per kilobase per million reads, scaled to background signal) from normoxic (red) and hypoxic (blue) MCF-7 cells are plotted for ± 10-kb flanking regions.

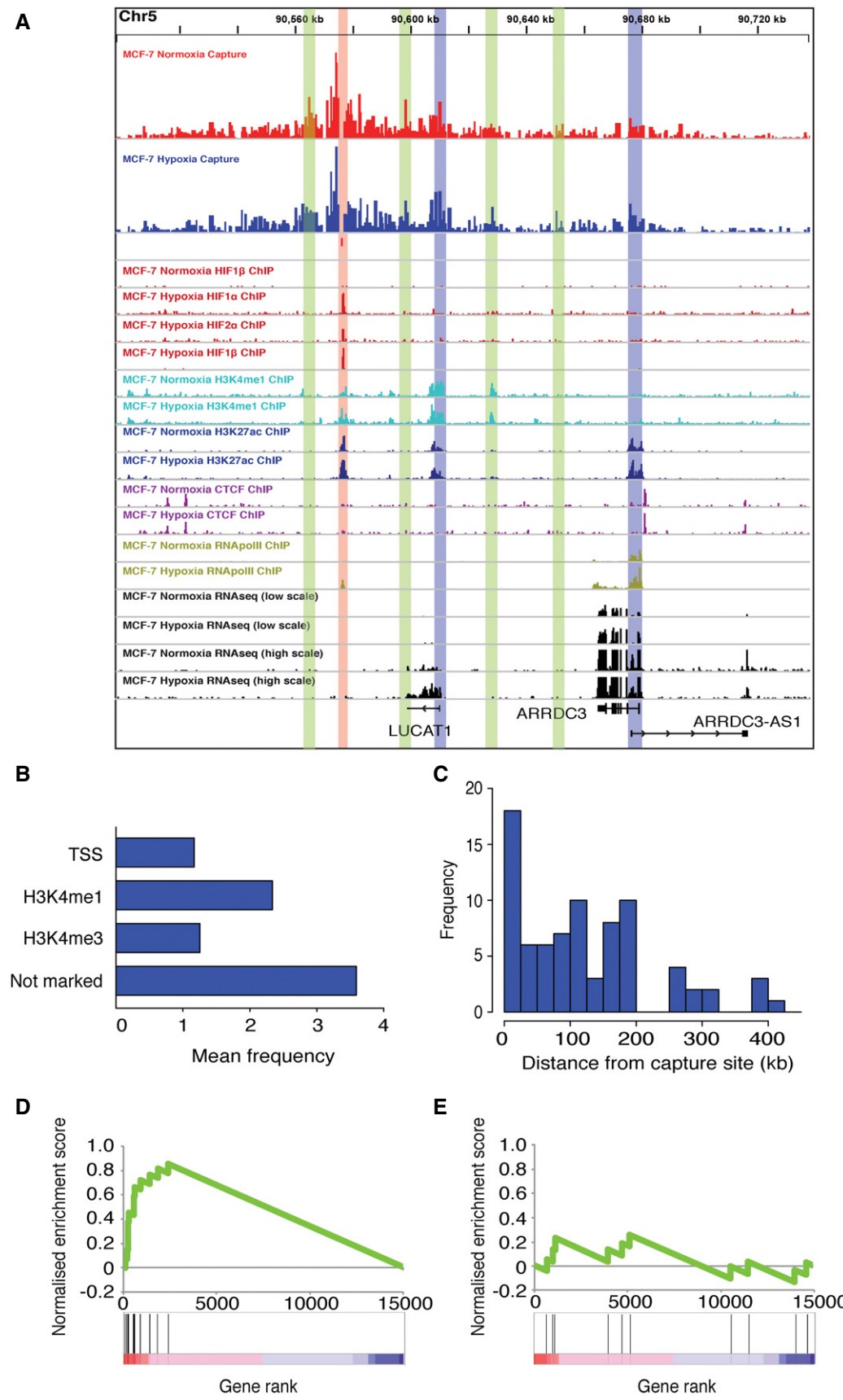

**Figure 2.**

**Figure 2.  Distant HIF-binding sites make physical contact with the promoters of HIF target genes.**

A   Capture-C from hypoxic MCF-7 cells (top) and ChIP-seq tracks for subunits of HIF, H3K4me1, H3K27ac, CTCF, and RNA-seq in normoxic and hypoxic MCF-7 cells as indicated at the ARRDC3 (arrestin domain containing 3) and LUCAT1 (lung cancer associated transcript 1, non-protein coding) gene loci. RNA-seq tracks are shown in both low scale and high scale to represent regulation of transcripts with differing transcript levels. Capture-C was performed using oligonucleotides to a "viewpoint" region at the HIF-binding site (highlighted in red). Interaction with the TSS of both LUCAT1 and ARRDC3 is highlighted in blue. Association with regions, including additional enhancers, is highlighted in green.

B   Regions associated with HIF-binding sites were classified according to the presence of a UCSC-annotated TSS, or coincidence with H3K4me1 or H3K4me3 ChIP-seq signal. The bar chart shows the average number of each type of site that interacted per HIF-binding site.

C   Histogram showing the frequency distribution of elements interacting with HIF-binding sites according to absolute distance from the "viewpoint" site on the horizontal axis.

D, E   Genes at which the TSS interacted with a HIF-binding site (D) and genes adjacent to HIF-binding sites, which did not interact with the site (E), were subjected to GSEA using a ranking metric combining fold-change in hypoxia and significance [25]. The ranking metric on the horizontal axis is shown as a color scale at the bottom. All the genes whose promoters were associated with HIF-binding sites appeared among the most strongly hypoxia-induced genes, while non-interacting genes showed no enrichment.

signal at these gene loci. This suggests that the observed chromatin–chromatin interactions are functional and that Capture-C may be used to predict the target promoter(s) of intergenic HIF binding elements, even when the genes lie at large distances and are not necessarily the closest locus to the HIF-binding element. It also provides clear evidence, across multiple gene loci that promoter-distant HIF-binding sites can physically interact with hypoxia-regulated gene promoters.

In addition to showing association with gene promoters, HIF-binding sites also interacted with an average of approximately three additional distant enhancer elements as characterized by the presence of H3K4me1 and H3K27ac (Fig 2B and Appendix Fig S2). This suggests more extensive convergence between HIF signaling and additional transcriptional regulatory pathways than is apparent from examining sequences local to the HIF-binding site.

### Chromatin interactions with HIF-regulated promoters

In addition to observing frequent weak ChIP-seq signal at sites interacting with a strong peak, several gene loci (e.g., NDRG1, DDIT4, ERRFI1, EGLN3) harbored multiple strong signals of roughly equal intensity, each of which contained a consensus HRE motif. These are among the most hypoxia-upregulated genes in MCF-7 cells (ranked 8, 28, 68, and 35, respectively) [4]. This raised the question as to whether the promoters of these genes are able to interact with more than one HIF-binding site. To answer this, we performed Capture-C using the promoters of these genes as "viewpoint" sites. This identified clear association between the promoters of these genes and multiple HIF-binding sites, both upstream and downstream of the promoter (Fig EV1). As multiple HIF-bound enhancers are brought into close proximity with the promoters of these genes, this indicates that more than one HIF-binding site may contribute to the regulation of a single gene, perhaps contributing to greater amplitude of regulation by hypoxia.

Glycolysis is one of the best-characterized HIF-regulated pathways, with at least one isoenzyme catalyzing each step of the pathway being a HIF target gene [4] (Appendix Fig S3). Unexpectedly, inspection of HIF ChIP-seq signals revealed that these gene loci bound both HIF-1 and HIF-2 with similar intensity, at or very close to their promoter, although all but one gene was exclusively regulated by HIF-1 (Fig EV2A–C). Interestingly, the exception, HK1, has a HIF-binding site within an intron of its gene, which although intragenic is (unusually for glycolytic genes) approximately 50 kb from the promoter (Appendix Fig S3). This raised questions as to whether functional selectivity among bound HIF isoforms might be a general

function of distance from a promoter. To test this, we interrogated our genomewide datasets [4,7]. This analysis revealed a clear difference between promoter-proximal and promoter-distal sites that bound both HIF isoforms. Promoters bound by both HIF-1 and HIF-2 were almost entirely upregulated by HIF-1 alone, whereas promoter-distant HIF-binding sites were found to be associated with genes that were upregulated by either or both HIF-1 and HIF-2 (Tables EV1 and EV2 and Fig EV3).

Since HIF binding at glycolytic gene loci is strongly promoter associated (Fig EV2C), we next determined the distribution of other enhancers at these loci. Despite the promoter-proximal nature of HIF binding, Capture-C, using the promoters of these genes as "viewpoint" sites, identified frequent interactions with more distant enhancers (Fig EV2D), indicating that the pattern of HIF binding observed does not merely reflect short-range interactions at these loci.

### HIF-binding sites are precomplexed with interacting elements before HIF binds

There are reports that certain inducible transcription factors bind to preexisting chromatin complexes [26–29], while others induce new patterns of interaction [30–33]. However, there have been very few studies that examine the question at high resolution and at scale. Since HIF-bound enhancers and HIF-regulated promoters bear active histone modifications prior to HIF stabilization, we next wished to determine whether and to what extent *cis*-interactions between these sites were pre-established or to what extent they were induced by HIF binding. We therefore performed Capture-C on MCF-7 cells, incubated in normoxia or hypoxia using the same capture probes to HIF-binding sites as above. The interaction frequencies in hypoxia correlated strongly with those in normoxia (Fig 3A and B). Only about 3% of *cis*-interactions were significantly altered in frequency following stabilization of HIF, and the magnitude of these changes was small (1.4- to 1.9-fold). Thus, induction of HIF does not greatly affect the pattern or strength of chromatin–chromatin interactions from HIF-bound sequences. To further assess the effect of HIF activation on chromatin structure, we also used ChIP-seq to examine binding of CTCF, a protein that has been shown to coordinate chromatin architecture [34]. Consistent with our Capture-C analysis, CTCF binding flanking HIF-binding sites was not significantly altered by hypoxia (Fig 3C).

To extend these observations, we next examined *cis*-interactions from HIF-binding sites in the renal cancer cell line 786-O. This

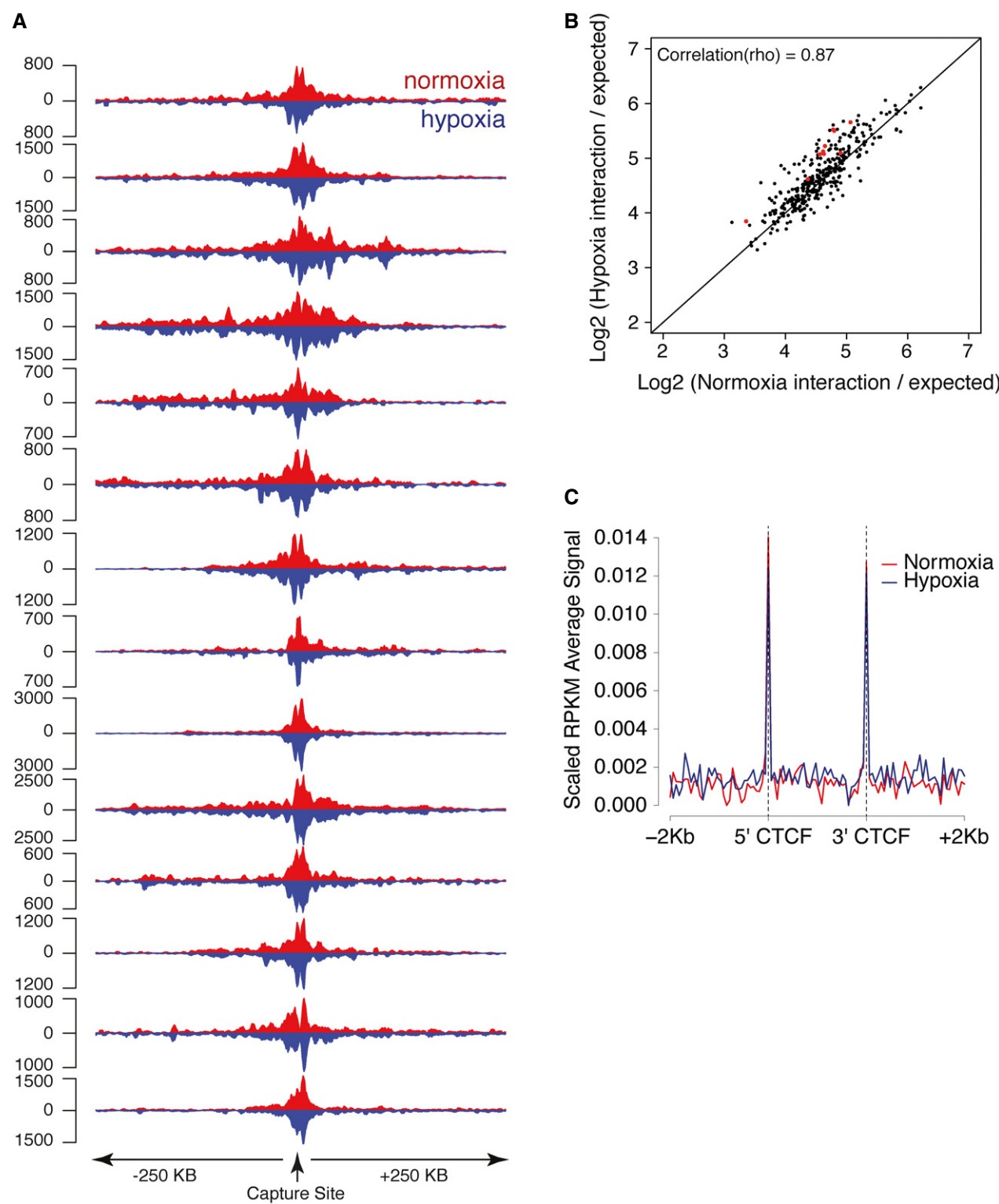

**Figure 3. Chromatin interactions with HIF-binding sites are pre-established in normoxia.**

A   Capture-C tracks (averaged across the two replicates) from normoxic (red) and hypoxic (inverted in blue) MCF-7 cells are shown for each HIF-binding site captured.
B   The interaction frequency (normalized to the number of informative reads in each dataset and to the expected distribution of counts at a given distance from the capture site) in hypoxia (vertical axis) was plotted against that in normoxia (horizontal axis) for each site that interacted significantly with a HIF-binding site in either normoxia or hypoxia or both. Highlighted in red are sites that had a significantly different interaction frequency in either condition.
C   CTCF ChIP-seq peaks immediately flanking the captured HIF-binding sites were identified, and the average normalized CTCF ChIP-seq signal (RPKM) was plotted for normoxic (red) and hypoxic (blue) MCF-7 cells.

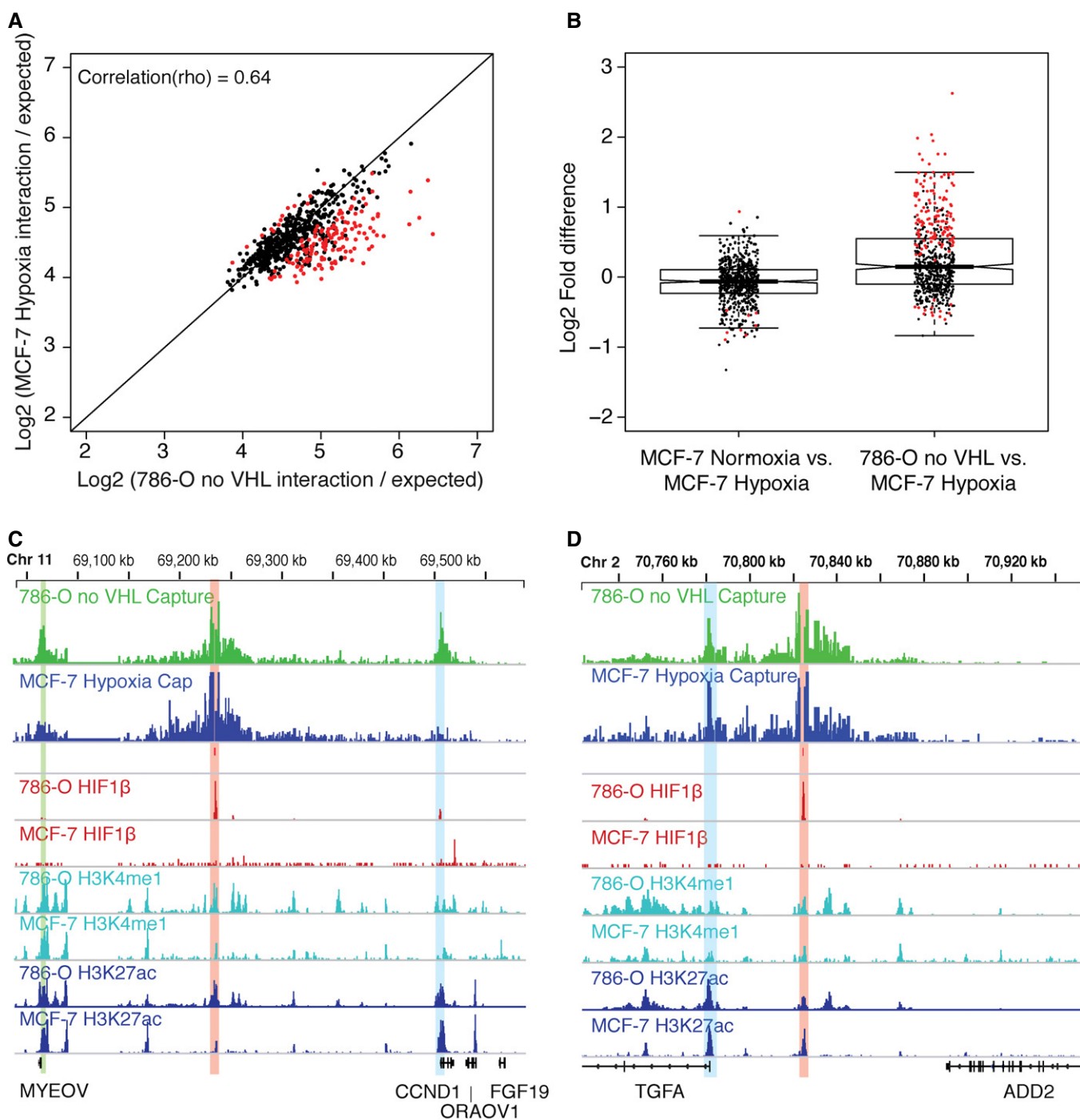

**Figure 4.  Cell-type specificity of chromatin interactions.**

A    The normalized interaction frequency (as above) in MCF-7 cells cultured in hypoxia (vertical axis) was plotted against that in VHL-defective 786-O cells in normoxia (horizontal axis) for each site that interacted significantly with a HIF-binding site in either cell line or both. Highlighted in red are sites that had a significantly different interaction frequency in either cell type.

B    Box and whisker plot showing the log2 (fold difference) between MCF-7 cells in normoxia and hypoxia (left) and VHL-defective 786-O cells in normoxia and MCF-7 cells in hypoxia (right) from Figure 3B and panel (A), respectively. Highlighted in red are sites that had a significantly different interaction frequency in either condition/cell type. Thick horizontal line denotes median. Boxes denote interquartile range and whiskers and thin bars denote 95% range.

C    Capture-C (MCF-7 hypoxia: blue; 786-O: green) at the CCND1 locus from an enhancer (red highlight) bound by HIF almost exclusively in VHL-defective 786-O and showing cell-specific *cis*-interactions, blue highlighted region indicates region of interaction at the TSS of CCND1, and green highlighted region indicates region of interaction at an enhancer 3′ to the MYEOV gene.

D    Capture-C (MCF-7 hypoxia: blue; 786-O: green) at the TGFA locus from an enhancer bound by HIF exclusively in VHL-defective 786-O cells but showing constitutive interaction. Blue highlighted region indicates interaction with the TSS of TGFA. Green highlighted region indicates interaction with an enhancer immediately 3′ to MYEOV.

cell line lacks functional pVHL, which is a common and early event, critical to the development of kidney cancer, and leads to long-term constitutive activation of HIF. To study the effect of short-term hypoxia in this context, wild-type VHL was stably reintroduced (786-O+VHL) and cells were studied in both normoxia and hypoxia. Again, we found chromatin interactions to be largely pre-established before hypoxic stabilization of HIF (Fig EV4). Finally, we examined whether long-term HIF stabilization resulting from VHL inactivation might alter *cis*-interactions with these HIF-binding sites by comparing Capture-C signals in VHL-reconstituted (786-O+VHL) and defective (786-O-VHL) 786-O cells (Appendix Fig S4). For the large part, interaction frequencies were comparable in VHL-reconstituted and VHL-deficient cells, with the exception of reduced interaction with the promoter of TSC22D2 in VHL-reconstituted cells versus VHL-defective cells, which correlates with a lack of hypoxic inducibility of this VHL-regulated gene in these cells [35,36]. Thus, neither short-term activation of HIF by hypoxia, nor its long-term stabilization by VHL inactivation greatly alters chromatin interactions with HIF-binding sites.

Despite the potential for both HIF and hypoxia to modify chromatin through multiple interactions with histone and chromatin-modifying proteins [10–12], neither short-term activation by hypoxia nor long-term activation in VHL-defective cells appears to substantially alter patterns of chromatin conformation local to HIF-binding sites, at least as identified in these assays and in these cancer cell lines. Taken together, this reveals a model of induction of the HIF pathway in which the chromatin structure between HIF-binding sites and HIF-activated promoters is present prior to HIF induction, distant active enhancer elements being precomplexed with target promoters that are themselves preloaded with paused RNApol2. The processes defining the chromatin structure in normoxic cells are unclear, but likely include DNA binding and structural proteins, including the cohesin complex, that are involved in the establishment and maintenance of the chromatin landscape within which HIF operates and which are thus important in determining the shape of the hypoxia response.

### Cell-type-dependent patterns of chromatin interaction with HIF-binding sites

Both the binding of HIF and its transcriptional targets vary considerably between cell types [37]. We therefore examined the extent to which these cell-type differences are reflected by differences in chromatin interactions by comparing Capture-C signals in hypoxic MCF-7 cells and in the VHL-defective 786-O cell line, in which HIF is constitutively stable (Fig 4A). This analysis showed that approximately 20% of *cis*-interacting sites differed significantly between the two cell types (1.2- to 6.0-fold), although many other *cis*-interactions were preserved. This result is in contradistinction to the roughly 3% of *cis*-interacting sites that differed between normoxia and hypoxia in MCF-7 cells (Figs 3B and 4B).

Sites that bound HIF in both cell lines interacted with the same promoter in each cell type, suggesting that the transcriptional target of a specific binding site may be independent of cell type. Conversely, sites that bound HIF in one cell line, but not the other (Appendix Table S1), showed a more heterogeneous picture of

*cis*-interactions (Fig EV5). In particular, significantly reduced association with the target promoter was observed, at three loci, in the cell type, which did not bind HIF at the "viewpoint" enhancer locus (versus that in the HIF-binding cell type; Fig 4C). However, at the remaining sites, *cis*-interaction with the target promoter was clearly preserved in the cell type that lacked HIF binding (Fig 4D). These findings support a model in which preexisting HIF-independent patterns of chromatin interaction define potential HIF targets, but also identify a second mechanism of selectivity, whereby additional factors contribute to cell-type-specific HIF binding at common preformed active chromatin complexes. The marked renal tissue specificity of pVHL-associated cancer [38] has focused attention on specific components of the HIF pathway that might be oncogenic in this context. Interestingly, two genes that have been defined as contributors to VHL-associated oncogenesis, CCND1 [39,40] and TGFA [39,41], are representative of each group: CCND1 among those manifesting cell-specific chromatin conformation (Fig 4C), and TGFA among those showing cell-specific HIF binding despite similar patterns of interaction (Fig 4D). Further work aimed at defining the mechanisms underlying these differences will be of interest.

In summary, the present work has revealed complex patterns of preexisting chromatin interaction between multiple HIF-binding enhancers and promoters that shape the transcriptional architecture of the HIF response. This supports a model in which binding of HIF to primed functional chromatin complexes may enable rapid activation of the HIF transcriptional program. Since HIF activates gene expression through release of promoter-proximal paused RNA polymerase 2 [4,6], this may be a strategy specific to this type of gene regulation, rather than a general mechanism for enhancer-induced transcriptional activation. It also distinguishes at least two other mechanisms that operate on top of these frameworks to define selective patterns of gene activation: cell-type-specific HIF binding to these preformed complexes and isoform-specific activity of the bound HIF.

## Materials and Methods

### Cell culture

MCF-7 and 786-O cells were cultured in Dulbecco's modified Eagle's medium (DMEM) supplemented with 10% fetal bovine serum, 2 mM L-glutamine, 100 U/ml penicillin, and 100 μg/ml streptomycin. For hypoxia treatment, cells were incubated for 16 h in an In Vivo2 Hypoxia Work station (Ruskinn Technology) at 0.5% oxygen.

### ChIP-seq

Chromatin immunoprecipitation was performed as previously described [7] using antibodies against HIF-1 (PM14), HIF-2 (PM9) [42,43], HIF-1 (Novus NB-100-110), H3K4me1 (Millipore, #07-436), H3K4me3 (#9751, Cell Signaling Technology), H3K27ac (#ab4729, Abcam), CTCF (#07-729, Millipore), and RNApol2 (#sc-899, Santa Cruz Biotechnology). Each ChIP was performed in duplicate. Libraries were prepared using PrepX Complete ILMN DNA library kit (#400076, Wafergen).

### Capture-C

Capture-C libraries were prepared as previously described [16]. Cells were fixed with 2% (vol/vol) formaldehyde for 10 min, quenched with 125 mM glycine in PBS, and then lysed in cold lysis buffer (10 mM Tris–HCl, pH 8, 10 mM NaCl, 0.2% IGEPAL, 1× complete protease inhibitor cocktail (Roche). Chromatin was digested with DpnII (New England Biolabs) at 37°C overnight. Fragments were then diluted and ligated with T4 DNA ligase (Thermo Scientific) at 16°C overnight. Cross-linking was reversed by overnight incubation at 60°C with proteinase K (Bioline). The 3C libraries were then purified by phenol–chloroform and chloroform extraction followed by precipitation in ethanol at −80°C overnight. Digestion efficiency was determined by qPCR and gel electrophoresis. Sequencing libraries were prepared from 5 μg of each 3C library by sonication using a S220 focused ultrasonicator (Covaris) to a average size of 200 bp and indexed using NEBnext reagents (New England Biolabs) according to the manufacturer's protocol. Enrichment of 1–2 μg of an indexed library incubated with 13 pmol of a pool of biotinylated oligonucleotides (Integrated DNA Technologies or Sigma-Aldrich; Appendix Table S1) was performed using the SeqCap EZ system (#06953212001, Roche/NimbleGen) following the manufacturer's instructions. Two rounds of capture employing 48–72 and 24 h hybridizations, respectively, were used. Capture enrichment was determined by qPCR. Correct library size was confirmed using a Bioanalyzer DNA 1000 or Tapestation D1000 kit (Agilent), and DNA concentrations were determined using a Qubit 2.0 Fluorometer (Thermo Fisher Scientific).

### High-throughput sequencing

All sequencing was performed on the HiSeq 2500 or HiSeq 4000 platforms using paired-end 50- or 75-bp protocols (Illumina).

### Accession codes

Capture-C data from this study are available from Gene Expression Omnibus at GSE78100 and ChIP-seq are available at GSE28352 (HIF-1α, HIF-2α, and HIF-1β ChIP-seq in MCF-7 cells), GSE32970 (ENCODE DNase-seq in MCF-7 cells), GSE67237 (HIF-2a and HIF-1b ChIP-seq in 786-O cells), GSM1011120 (FAIRE-seq in 786-O cells), and from EMBL-EBI Array Express at E-MTAB-1994 and E-MTAB-1995 (RNApol2 and H3K4me3 ChIP-seq and RNA-seq in MCF-7 cells) and GSE78113 (other histone modifications and CTCF ChIP-seq).

### ChIP-seq analysis

ChIP-seq data were processed as previously described [36]; peaks were called using MACS [44] and TPIC [45]. ChIP reads were counted under peaks and were normalized to the total library size and the width of the peak using RPKM [36].

### Capture-C analysis

Reads were processed, *in silico* digested, and aligned to the human genome using Bowtie 1.0.0 (http://bowtie-bio.sourceforge.net/index.shtml). Interaction frequencies were determined using

CCanalyser2.pl (https://github.com/telenius/captureC/releases) as previously described [16] (Appendix Table S2). The presence of undigested self-circles was examined using the method of Jin *et al* [26]. Only small undigested self-circles of < 2 kb were detected, likely reflecting the higher cutting frequency of DpnII. Beyond this distance, the ratios of outward/same-strand and inward/same-strand reads were both 0.5, consistent with efficient cutting and re-ligation of DpnII fragments (Appendix Fig S5).

### Identifying significant interaction regions

#### Normalization of observed counts
To identify regions of significant interaction with the capture probe, we followed an approach similar to that used previously [26].

Firstly, the observed interaction count $Y_{x,j}$ between a DpnII fragment $x$ and a specific capture probe $j$ was normalized to the total captured interactions for the capture probe $Y_j$ and to the size of the DpnII fragment $W_x$.

$$\bar{Y}_{x,j} = \frac{Y_{x,j}}{Y_j W_x} NF$$

where $\bar{Y}_{x,j}$ is the normalized count, and $NF$ is a numerical normalization factor, $10^8$, to adjust the counts to a similar mean before normalization.

#### Empirical estimation of random background interaction counts
The random interaction rate between the capture fragment and nearby chromatin is expected to decrease exponentially with distance. Accordingly, we used the normalized interaction counts across all experiments to empirically estimate the expected background distribution of interaction counts at a given distance.

The region surrounding each bait site, $j$ was divided into 2-kb bins and the normalized count for each 2-kb bin estimated by summing the counts for each DpnII fragment, $x$ within bin $i$.

$$\bar{Y}_{i,j} = \sum_{x=1}^{N} \bar{Y}_{x,i,j}$$

where $N$ is the number of DpnII fragments within each 2-kb bin $i$.

The expected interaction frequency at bin $i$ is then given by the interaction counts in bin $i$ averaged across all the probes $j$ for every experiment in that cell type (Appendix Fig S6A):

$$E[\bar{Y}_i] = \frac{\sum_j \bar{Y}_{i,j}}{N_j}$$

where $N_j$ is the number of bait sites.

However, background counts may vary from experiment to experiment, so the average interaction counts for a given cell type were then corrected using a scalar correction factor $C_e$ derived for each experiment, $e$:

$$E[\bar{Y}_{i,e}] = E[\bar{Y}_i] * C_e$$

using a method similar to that used in DESeq [46], where:

$$C_e = \text{median}\left(\frac{G\bar{Y}_{e,i}}{E[\bar{Y}_i]}\right)$$

and where $G\bar{Y}_{e,i}$ is the geometric mean of all the interaction counts across all the probes $j$ for experiment $e$:

$$G\bar{Y}_{e,i} = \left(\prod_{j=1}^{N_j} \bar{Y}_{e,i,j}\right)^{1/N_j}$$

Similarly, a normalization factor, $C_j$, for each capture probe was calculated to normalize the capture probe efficiency:

$$C_j = \text{median}\left(\frac{G\bar{Y}_{e,j}}{G\bar{Y}_e}\right)$$

where $G\bar{Y}_{e,j}$ is the geometric mean of the probe across all experiments for the same cell line, averaging over all the bins:

$$G\bar{Y}_{e,j} = \left(\prod_{j=1}^{N_i} \bar{Y}_{e,i,j}\right)^{1/N_i}$$

and $G\bar{Y}_e$ is the geometric mean of all the interaction counts across all the experiments in the same cell line:

$$G\bar{Y}_e = \left(\prod_{j=1}^{N_i N_j} \bar{Y}_{e,i,j}\right)^{1/(N_i * N_j)}$$

### Log–log model of interaction counts

To model the interaction count at a given distance from the capture site, we then fitted a log–log power law for the relationship between the expected interaction count $E[\bar{Y}_i]$ in bin $i$ and its distance from the center [26] (Appendix Fig S6B) in a certain cell line. We then regressed $E[\bar{Y}_i]$ against the distance and defined $\mu_i$ as the randomly expected interaction count at distance $d_i$ as follows:

$$\mu_i = \frac{e^k - 1}{d_i^{\delta}}$$

where $e^k - 1$ is the expected signal closest to the capture site (i.e., at bin 1) or the intercept, and $\delta$ is the exponential decay parameter for the interaction signal. Using this model, $\delta$ was then calculated, for every cell line used, using the maximum likelihood estimate.

The distribution of interaction counts, $E[\bar{Y}_i]$, at bin $i$ for any capture probe has been shown to follow a negative binomial distribution, which allows for fitting an extra parameter $\beta_i$ for the mean–variance relationship:

$$E[\bar{Y}_i] \sim NB\left(r_i = \frac{\mu_i}{1 - \beta_i}, P_{i,j} = \frac{\beta_i - 1}{\beta_i}\right)$$

$$var(E[\bar{Y}_i]) = \beta_i \mu_i$$

$\beta_i$ was previously observed, to be fixed [26]; however, we found that this parameter is a linear function of the mean counts (Appendix Fig S6C) [47]:

$$\beta_i = \alpha \mu_i + k$$

Accordingly, we used generalized linear modeling to empirically estimate the maximum likelihood parameters for $\alpha$ and $k$. This identified the expected distribution of counts at a given distance from the capture site as shown in Appendix Fig S6D.

### Calculation of interaction significance

Having built a model of the null hypothesis distribution at every bin $i$, we then used the observed interaction counts to test its significance.

We first segmented the interaction space around the capture probe, into overlapping windows of width 2 kb at an equally spaced distance of 200 bp. We then calculated the interaction counts $\bar{Y}_{i,j}$ as shown before. Finally, we normalized the interaction counts for every probe to a similar mean, to avoid any false positives:

$$\bar{X}_{i,j} = \frac{\bar{Y}_{i,j}}{C_j}$$

We then used the model built before to build the null hypothesis model at the observed distance of the window, to estimate the *P*-value. *P*-values were calculated using the cumulative distribution function (CDF) for the expected negative binomial distribution:

$$\rho(\bar{X}_{i,j}) = 1 - \sum_{\forall Y \le X} P(\bar{X}_{i,j}|NB(\mu_i, \beta_i))$$

### False discovery rate (FDR) correction

False discovery correction assumes an equal probability of significance across all the tests being conducted. However, within the same capture probe, this is a false assumption, since it is harder to call significance closer to the capture site, and at extreme distances and the distribution of *P*-values at different bins will therefore differ. Accordingly, stratified FDR correction [48], correcting at every bin across all the probes being captured, was used. This is justified, since all the probes are captured in one experiment.

### GC content and mappability

The effect of potential confounders, including GC content and mappability, was determined using previously described methods [26]. The GC content and mappability of statistically significant *cis*-interaction regions was comparable to that of control, non-*cis*-interacting, regulatory regions within 500 kb of bait sites (Appendix Fig S7).

### Differential significance between conditions

To perform differential analysis, we constructed a union of the significantly interacting regions across the two conditions (i.e., normoxia and hypoxia in MCF-7). We used any significant region that was called significant in any replicate. We then used an approach similar to that of DESeq [46].

Essentially, we used the likelihood ratio test (LRT) to examine the null hypothesis that the interaction frequencies at each site were similar across the two conditions. Specifically, a generalized linear model (GLM) with a negative binomial response was applied under the null hypothesis and compared to an unrestricted negative binomial GLM. The chi-squared test with 1 degree of freedom was then used to find the significance of each interaction, and *P*-values were adjusted for false discovery rate (FDR).

To test across different cell lines, we normalized the cell line mean interaction frequency. Accordingly, we used a similar normalization approach to the above: we calculated a correction factor for the interaction counts that would ensure that the median interaction counts across the cell lines being tested were similar:

$$C_{cl} = \text{median}\left(\frac{G\bar{Y}_{cl,j}}{G\bar{Y}_j}\right)$$

where $GY_{cl,j}$ is the geometric mean across probe $j$ in one cell line, and $GY_j$ is the geometric mean across probe $j$ for all the cell lines. $C_{cl}$ is then the median of this normalization factor for the cell line $cl$. The interaction counts for each cell line are then obtained using:

$$\overline{XC}_{i,j} = \frac{\bar{X}_{i,j}}{C_{cl}}$$

**Expanded View** for this article is available online.

## Acknowledgements

We thank Douglas Higgs for his help and advice with establishing the Capture-C assay in our laboratory and Angie Green and the High-Throughput Genomics Group at the Wellcome Trust Centre for Human Genetics (funded by the Wellcome Trust—090532/Z/09/Z) for the generation of the sequencing data. This work was supported by a Cancer Research UK (grant number A16016), the Ludwig Institute for Cancer Research, the Higher Education Funding Council for England, the Wellcome Trust (grant numbers 078333/Z/05/Z, WT091857MA) and the Deanship of Scientific Research (DSR), King Abdulaziz University, Ministry of Higher Education for Saudi Arabia.

## Author contributions

JLP, RS, JRH, PJR, and DRM designed the experiments. JLP, JS, and HC performed the experiments. JLP, RS, JRH, JOJD, and DRM performed data analysis. JLP, PJR, and DRM wrote the manuscript.

## Conflict of interest

PJR is a scientific co-founder and holds equity in ReOx Ltd., a university spin-out company that seeks to develop inhibitors of the HIF hydroxylases.

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
