## [Review Process File · EMBO Reports]

Manuscript EMBO-2016-42198

Capture-C reveals preformed chromatin interactions between HIF-binding sites and distant promoters

James Platt, Rafik Salama, James Smythies, Hani Choudhry, James Davies, Jim Hughes, Peter Ratcliffe, and David Mole

Corresponding author: David Mole, University of Oxford

Review timeline:

Submission date:	17 February 2016
Editorial Decision:	21 March 2016
Revision received:	20 June 2016
Accepted:	15 July 2016

Editor: Esther Schnapp

Transaction Report:

1st Editorial Decision

21 March 2016

Thank you for the submission of your manuscript to our journal. I am sorry for the slight delay in getting back to you; we have now received the full set of referee reports on your study that is pasted below, as well as referee cross-comments.

As you will see, the referees acknowledge that the findings are potentially interesting. However, they also point out that it is unclear in several cases whether the Capture C data are derived from normoxic or hypoxic conditions, whether HIF is already present at the chromatin loops in normoxia, and what happens at genes that are off or not bound by HIF in normoxia but induced by HIF in hypoxia. These are the most crucial concerns that must be addressed. The referees also note that several conclusions are not supported by the data and need to be toned down, and that more details about the experiments need to be provided. I think that the use of cancer cell lines (referee 2) is less of a concern, and that it is out of the scope of this study to investigate how the chromatin loops are made during normoxia (referee 3), although this would be interesting information of course.

Given these constructive comments, we would like to invite you to revise your manuscript with the understanding that the referee concerns must be fully addressed and their suggestions taken on board. Please address all referee concerns in a complete point-by-point response. Acceptance of the manuscript will depend on a positive outcome of a second round of review. It is EMBO reports policy to allow a single round of revision only and acceptance or rejection of the manuscript will therefore depend on the completeness of your responses included in the next, final version of the manuscript.

Revised manuscripts should be submitted within three months of a request for revision; they will

otherwise be treated as new submissions. Please contact us if a 3-months time frame is not sufficient for the revisions so that we can discuss this further. You can either publish the study as a short report or as a full article. For short reports, the revised manuscript should not exceed 25,000 characters (including spaces but excluding materials & methods and references) and 5 main plus 5 expanded view figures. The results and discussion sections must further be combined, which will help to shorten the manuscript text by eliminating some redundancy that is inevitable when discussing the same experiments twice. For a normal article there are no length limitations, but it should have more than 5 main figures and the results and discussion sections must be separate. In both cases, the entire materials and methods must be included in the main manuscript file.

Please remember to include all relevant information on data quantification and statistics in the figure legends.

We now strongly encourage the publication of original source data with the aim of making primary data more accessible and transparent to the reader. The source data will be published in a separate source data file online along with the accepted manuscript and will be linked to the relevant figure. If you would like to use this opportunity, please submit the source data (for example scans of entire gels or blots, data points of graphs in an excel sheet, additional images, etc.) of your key experiments together with the revised manuscript. Please include size markers for scans of entire gels, label the scans with figure and panel number, and send one PDF file per figure or per figure panel.

I look forward to seeing a revised version of your manuscript when it is ready. Please let me know if you have questions or comments regarding the revision.

REFEREE REPORTS

Referee #1:

Here, Platt and colleague investigate HIF binding at distance sites of chromatin, using a variant of chromatin capture, with focus on HIF binding sites. They demonstrate that HIF binding, and chromatin loop does not significantly change with hypoxia and that these sites are already marked for activation. This is a novel and interesting study, that clearly shows that for the majority of the HIF binding sites in intergenic regions, no big changes in chromatin structure are required. I just have a few points that would require some clarification.

From what is said, this analysis is based on 18 intergenic regions, the authors should which state the percentage of HIF targets are regulated in such a manner. Since HIF has hundreds to thousands of binding sites.

While it is very clear that the data supports the authors claims, the authors also state that these genes are already transcribed in normoxia, what would happen in a gene that is on/off such as CA9 for example?

The findings between HIF-1 and HIF-2 regulation of genes through their localisation at promoters or enhancer is very novel and interesting and should be placed in the main article, including Figure S6, and a list of the name of the genes analysed provided as supporting data.

Finally, just have a query as to the choice of 786-O cells for the analysis, since they only have HIF-2 and HIF-1. Sure, something like RCC4 would have been better to compare to MCF-7 that also has both isoforms, was there a reason for this? It might have changed the results?

Referee #2:

In this paper, Platt et al. show that the majority of the epigenetic marks and chromatin loops surrounding hypoxia inducible genes are pre-established in normoxic conditions and that the activation of hypoxia-responsive transcription factor HIF does not cause dramatic changes in 3D genome organization. They conclude that HIF must be acting through already pre-existing genome

structure to activate target genes, but that these existing loops can be used to determine which of numerous distal gene targets will be regulated by HIF in hypoxia. The approach, data, and analyses in this paper are mostly acceptable, and the question of how the genome organizes its response to this kind of signal is interesting, though numerous previous studies have looked at changes or lack of change in chromatin looping in other signal response systems in the past. Despite these previous studies, there is still some particular interest in examining changes in looping during hypoxia since chromatin structure changes have been implicated in this response in previous work. Also, though other papers have already shown the principle that quick signaling responses often do not involve changes in genome structure, this is still not accepted as "well known", so other evidence is useful.

All this being said, there are numerous points that require addressing to make sure that the authors are interpreting their data correctly and that their data support their conclusions. The authors make some claims (detailed below) that are too strong for their data or which make too many unsubstantiated assumptions. My general concern is that the authors too freely assume that a high contact frequency in their data is the same thing as a specific "loop" (when some interactions are just part of a broader domain structure) and that a loop would mean direct regulation (when other evidence would be needed to show that a looping interaction causes regulation). The authors also need to do a better job of quantifying their comparisons (described in more detail below) rather than using either only visual inspection, anecdotal examples, or the strength correlation of called peaks. A more rigorous comparison is needed: can the authors compare how many peaks change between replicates vs. across conditions? If the changes are about the same, then indeed, nothing changes in hypoxia. Actual numbers of peaks that change in hypoxia or across cell types should be quantified. Further, in order to make the argument that these interaction patterns show "little change", it would be best to have an example of a case where similar profiles show a "substantial change". Can other literature be cited for this purpose? I am concerned that the differences in earlier work, where some papers show little change in interaction patterns and others do show changes, reflect more the pre-existing bias or subjective interpretation of different authors more than difference in data. So, the lack of difference claimed here needs to be carefully evaluated.

The other major concern I have about the paper is the use of cancer cell lines. It is known that cancer cells vs. non-cancer cells respond differently to hypoxia, and that cancer cells often have to adapt to hypoxic tumor environments. So, it concerns me that the "pre-existing loops and histone marks" might be specific to the cancer cells studied. Can the authors provide evidence that primary cells also have pre-existing epigenetic marks poising them for hypoxic response?

Also, just as a general point, in order to make this paper fit the mission of EMBO Reports (a single clear message), the authors should focus on supporting their main conclusion (that pre-existing looping defines potential target genes of HIF sites and the hypoxia response makes little change in this looping structure) and not get distracted by lots of other interesting data like whether some genes are regulated by either or both HIF1a and HIF1b. If these other points can be developed into something conclusive and interesting, then perhaps a different journal is appropriate, but in the current manuscript, they are a bit distracting (And, as I list below, some of the associated data is questionable).

Specific Comments:

- 1) The Title (and claims elsewhere in the paper) make it sound like HIF only binds at locations where looped contacts to genes occur, while the data only show that many of the tested HIF sites have interactions with genes. This cannot be extrapolated to claim that all 1000 HIF sites in the genome loop to genes, or to prove that these interactions are regulatory. These claims should be reworded.
- 2) The statement in the introduction that "most intergenic HIFbinding regions do indeed function as transcriptional enhancers" is too strong. You say 9 of 14 with good signal interacted with at least 1 TSS (which were probably already pre-selected for being in good candidate regions) This does not cover the 100s or 1000s of sites genome wide. Nor does it show these actually function as enhancers.
- 3) The use of several qualifying phrases in the text seems unsupported. For example, on Page 3, how do you define a "surprisingly large" number of sites in an intergenic region? More than would be

expected at random? Surprising only if we think no long distance regulation happens? More than for other TFs? Similarly, the statement that there are "very extensive cis-acting interactions" (page 4) is hard to justify: How do you define "very extensive"? More than a random regions? Or more than other non hypoxic TFs during this response?

4) When the authors mention (Page 4) that hypoxia has been reported to affect genome structure, they should cite Kirmes et al., 2015 as a recent study on this topic.

5) The authors state that Fig 1 shows that HIF binding occurs at enhancers that are already partially active in normoxic cells. But this statement comes from the average signal. Are there actually separable classes of enhancers, some of which are already active and some of which are not at all? What is the distribution around the averages shown in Fig 1? If enhancer regions are called in each condition, how many of them overlap between normoxia and hypoxia.

6) When the authors (page 6) speak of selecting HIF regions that "bore enhancer marks", are they speaking of normoxia enhancer marks? Hypoxia enhancer marks? A good negative control for their conclusions might be to do Capture C from one region that did not already have enhancer marks.

7) Page 7 (top): what are "functional domains"? Do the authors mean TADs? It would be more useful to evaluate how many interactions are within TADs, rather than within an arbitrary 500 kb.

8) Page 8 (top): "HIF can effect transcriptional regulation by looping..." This claim seems too strong: To prove this you would have to show that removing the particular enhancer would affect transcriptional regulation.

9) Page 8 (middle): "suggests more extensive cooperativity between HIF and other transcriptional regulatory pathways" Maybe-but it could be alternately that all these other interactions are pre-set up for totally independent pathways.

10) Page 11: The authors mention the "chromatin structure required for HIF activated gene expression", but they haven't directly shown that chromatin structure is required for HIF activated gene expression. It is possible that you could disrupt the structure and still see activation-- it is not proven otherwise here.

11) Page 12: it is never indicated which sites are bound by HIF in both cell lines. Should this in a supplemental figure?

12) The finding about 3 genes looping to a co-regulated coding and non-coding gene pairing is quite interesting, but anecdotal and perhaps distracting from the main message of the paper

13) A major question that is not addressed in this paper is: How do the interaction patterns of these HIF sites relate to other nearby non HIF sites? Would any given captured region closely interact with all these marked regions? Or are these loops really specific to HIF sites? The authors could address this by trying a negative control capture of some non HIF site within the general region of the genes and elements of interest. This would help address the concern that contacts might be general structural patterns rather than specific "loops" per-se.

Regarding the Methods section:

14) Generally, the analysis and normalization of the Capture-C data is thoughtful and solid. But, I am concerned about the use of "candidate interacting regions" based on epigenetic marks as a prior probability. It is a bit unclear in the methods, but it seems that they only considered candidates by epigenetic marks as possible regions of interaction. How can they then robustly draw conclusions about whether other looping interactions (that lack epigenetic marks) might change in hypoxia? The initial assumption that only marked regions can interact is not really proven in the literature. The authors should also do a comparison of all candidate interacting regions when they are comparing normoxia to hypoxia, and then focus on marked regions if appropriate.

15) How do the authors deal with a difference in sequencing coverage that may occur at certain locations in the genome due to a different frequency of restriction sites, different mappability, etc.?

16) Does the analysis approach for identifying pairs of reads eliminate undigested self-circles? (these will appear to be interactions > the 1 kb exclusion distance but might actually be the product of undigested self-circles?) Previous literature (ref 27 in this paper, Jin et al) shows that these artifacts can be prominent out to 5-10 kb. This may not be true for DpnII, but the authors should explicitly check for potential self circle pairs.

17) "Normalized interaction frequency" is used in figures and referenced in the Methods without being carefully defined. Does this mean normalized by the distance decay? By the total number of reads in the dataset?

18) The manuscript indicates that CaptureC data are currently being submitted to GEO. This must be completed by publication date.

19) I do not see a record of what capture probes were used. This must be included.

20) A table of all determined interacting elements (FDR 5%) should be provided (with GEO or as supplement)

Figure 1:

21) Define better in the figure legend or methods where your datasets come from (all but CHIP-Seq pre-published, right?) What does "scaled" mean in the y axis? Define a bit better somewhere.

Figure 2:

22) It is unclear here and in a few other figures whether the Capture C shown here was done in hypoxia? Normoxia? Is the HIF binding from normoxia or hypoxia? (I assume hypoxia, else the paper interpretations make no sense, but it should be indicated)

23) 2D: label x axis "gene rank for hypoxia induction" (or similar) What does color scale on x axis mean? (ie are all pink/red induced and all blue repressed?) This is important to support the conclusion in the text that target promoters were "invariably" hypoxia-upregulated (that is only true if light pink is truly significantly upregulated)

24) E: the authors say: "genes closest to HIF-binding sites, which did not loop to the site" Does this mean adjacent genes that did not loop? Or not necessarily adjacent but within a certain distance? In these cases, do these TSS lack active marks?

Figure 3:

25) I don't understand the point of Fig 3D. Why look at CTCF within TADs vs. everywhere? Conclusion from this is not explained.

Figure 4:

26) What are the highlighted genes? Known cell type specific ones? Is MYEOV not a cell type specific gene? Hypoxia induced? It is also cell type specific loop-why?

Fig S1:

27) Reproducibility, here and elsewhere, is measured by interaction frequency comparison for "looping sites." The authors should check whether the relative levels of interaction among these sites are mostly dominated by distance-dependent decay. (that is, the higher values are just closer to the capture site than lower values?) If the dynamic range of these measurements is mostly distance dependent, then the fact that these interaction frequencies stay the same across conditions or replicates is not surprising. Check this by plotting correlation of interactions normalized by expected at distance, for example. Also, it would be very useful to report what percentage of called interactions in each replicate overlapped. This is true any time the authors compare two Capture C datasets: there needs to be a measure of how many peaks overlapped, not only how much shared peaks are similar in frequency.

Fig S3:

28) What Chromosome is this? (state at top of figure) It appears that there is actually a decrease in peaks in hypoxia in 2 cases?? (where HIF binds?) The authors need to indicate more carefully whether peaks were called significant in both normoxia and hypoxia. For this and other figures

showing Capture C data, is the data presented the result of pooled replicates?

29) Fig S5 C and D need scale indicators-0 to 5 what? Kb? The authors are showing that distant enhancers are interacting even though HIF sites are proximal- this point would be even stronger if the authors checked if this is the same distribution of interacting enhancers distances that they saw for other promoter captures in this paper.

Fig S5 and S6:

30) Can the authors provide evidence that the HIF2a knockdown is working given that it usually has no effect in Fig S5?

31) All these knockdown heatmaps should be clustered to show different categories of single and combined effects. Right now, it is very hard to interpret the jumble of different responses.

32) It is worrisome that some of the controls have such a variable Z score. (S6B in particular) I would assume that the control knockdowns should all have the same effect. How can the effect of a knockdown be known if the control is already affected?

Discussion:

33) Can the authors speculate about how enhancers are pre-looped to the "correct" promoters (not the ones that aren't to be induced). There must be some other (HIF independent) mechanism of specifying poised promoters?

34) The comments about 6,000 unique ligations, etc should move to the results section from the discussion.

35) It is a bit unfair to claim in the discussion that all but two HIF sites made physical contact with promoters when two of those were not actually called significant. Either the calling method has meaning and should be used consistently, or it doesn't and isn't useful.

36) If the genome architecture is all pre-defined, how can characterizing it reflect the "HIF transcriptional response" (as claimed on page 14)? Doesn't it more accurately determine the landscape that the HIF response can work with?

Referee #3:

The manuscript addresses the mechanisms of transcriptional activation by HIF using Capture C to analyze interactions between enhancers and promoters under normoxia and hypoxia conditions. The conclusions offer important insights into the process by which paused genes release into productive elongation during activation. The following are some comments that should help the authors improve the manuscript:

1. On page 5, the authors describe that "ChIP-seq was performed in MCF-7 cells, incubated in normoxia or hypoxia (0.5% oxygen, 16 hr), using antibodies to H3K4me1 and H3K27ac and analyzed together with RNA-seq and H3K4me3, RNAPol2 and HIF ChIP-seq datasets obtained under similar conditions". This gives the impression that the distribution of HIF was also examined under normoxia conditions, but I could not find this information in the manuscript. For example, in Figure 1A, are the promoter-distal HIF binding sites defined under normoxia or hypoxia conditions? I went back to previous manuscripts by the same authors where they published ChIP-seq data and this information is never given in any of these manuscripts. I am not an expert on HIF expression but I found several published papers indicating that HIF is transcribed under normoxia conditions, although the protein levels are regulated at posttranscriptional steps. This is a very important issue because it directly affects the main claim of the manuscript i.e. that HIF is recruited to preformed loops in hypoxia. However, if there is some HIF already present in cells under normoxia, then this claim may not be correct. The authors need to clarify this point and, in the process, write more descriptive figure legends.

2. Page 5. "These promoter-distal HIF-binding sites were highly accessible compared to promoters, and were strongly enriched for the enhancer mark H3K4me1, but had low levels of the promoter mark, H3K4me3". I think it is well established that strong enhancers transcribe high levels of eRNAs and also contain H3K4me3.

3. Page 5. "However, H3K27ac and RNAPol2 were also present at these sites in normoxia before HIF binds". As described under #1 above, it would be nice to show proof that this is true by showing

ChIP-seq data with no HIF signal at these sites. Even if levels of HIF are low in normoxia as determined by Western analysis, it has been shown in various publications that when levels of a DNA binding protein are reduced by 90% using RNAi. The actual amount of protein bound to DNA does not change dramatically. I think this point has to be clearly addressed because it is the main conclusion of the manuscript.

4. Page 6. "Capture-C allows an unbiased determination of distant interacting elements". Every experimental approach in biology is subject to biases.

5. Figure 2A. The authors should discuss the fact that LUCAT1 does not appear to be expressed in spite of a strong interaction with the HIF site used as bait. In addition, the authors should show the structure of the ARRDC3 and ARRDC-AS1 in an expanded view so that each gene can be visualized separately.

6. Page 6 and Figure 2A. The Capture C experiments displayed in this figure seem to suggest that there are additional interactions that appear quite strong based on the visual appearance of the data but were not called as significant by the computational approach used. I wonder if this could be due to the binning in 5 kb bins. Since the authors used DpnII to make the libraries, in principle they could use smaller bins if they had sufficient numbers of reads. I could not find information in the manuscript on how many QC'd reads were obtained for each experiment. This information should be presented in a supplemental table and discussed in the context of why interactions between HIF sites and promoters could not be detected in some cases (page 7).

7. Figure 3A. It is very difficult to appreciate the significance of the data presented in this figure. It appears that there are not significant changes in regions far from the bait. I wonder if it would be more informative to focus on the region surrounding the bait and try to examine interactions at high resolution i.e. either single fragment or 1 kb bins.

8. Page 10. : Neither CTCF binding close to HIF-binding sites, nor pan-genomic patterns of CTCF binding were significantly altered by hypoxia, suggesting that at least under these conditions, chromatin looping is not altered by induction of HIF (Figure 3C & D)". This is a very strong statement in the absence of any data to support it. Were the CTCF ChIP-seq experiments done in duplicate? How as the data analyzed to conclude that no alterations in CTCF binding were observed under hypoxia? It is possible that binding of CTCF is not affected but the distribution of cohesin is, and this regulates changes in looping.

9. Figure 4C. There seems to be at least one significant interaction in 786-O cells not present in MCF-7 cells that is not highlighted in the figure.

10. The conclusions of the manuscript would be more significant if the authors could address the issue of what is making the loops during normoxia. Cohesin and Mediator are two obvious candidates. Also, I feel that the authors do not emphasize sufficiently the fact that HIF-induced genes are paused, and that activation of these genes by recruitment of HIF to previously formed loops may be a strategy employed to release Pol II from paused genes, rather than a general mechanism for enhancer-induced transcription activation.

1st Revision - authors' response

20 June 2016

We are grateful to you for considering our revised manuscript EMBOR-2016-42198V2 '**Capture-C reveals preformed chromatin interactions between HIF-binding sites and distant promoters**' for publication in EMBO reports. We note that the referees found the paper of interest and feel that the new analyses strengthen our conclusions.

We attach a detailed point-by-point response to the concerns raised by the referees. In particular, we provide new analyses of both normoxic and hypoxic HIF binding to show that there is almost no detectable HIF-1beta signal in normoxia, when chromatin interactions are already present. We also clarify the conditions under which each analysis was performed, and where available include data for both normoxic and hypoxic cells. With very few exceptions, hypoxia-regulated genes are expressed to some degree in normoxia, despite the absence of detectable HIF binding. Nevertheless,

performed interaction between the promoter and HIF-binding sites was seen at some of the most highly regulated genes (e.g. NDRG1 – 8th biggest fold induction), although was not possible to determine for CA9, since the HIF-binding site is at the promoter.

We have included many of the extra analyses and controls requested by referee 2, which although they do not alter the conclusions reached in the paper, do increase the rigor of the findings. We also accept that several of our conclusions needed to be toned down and have done so. We have briefly discussed our choice of cell lines and how chromatin interactions might be established in normoxia, but agree with the editorial perspective on these issues. We have deposited our original source data with Gene Expression Omnibus and have included accession numbers in the manuscript. Finally, in the light of conflicting opinions from the referees regarding supplemental datasets, we have left these in Supplementary Information and left the manuscript formatted as a short report. However, we would be happy to be guided by editorial opinion on this.

Referee #1:

Here, Platt and colleague investigate HIF binding at distance sites of chromatin, using a variant of chromatin capture, with focus on HIF binding sites. They demonstrate that HIF binding, and chromatin loop does not significantly change with hypoxia and that these sites are already marked for activation. This is a novel and interesting study that clearly shows that for the majority of the HIF binding sites in intergenic regions, no big changes in chromatin structure are required. I just have a few points that would require some clarification.

From what is said, this analysis is based on 18 intergenic regions, the authors should state the percentage of HIF targets that are regulated in such a manner. Since HIF has hundreds to thousands of binding sites.

32% of HIF-1 & 49% of HIF-2 binding-sites are more than 10 kb away from their nearest annotated gene. We have added this information to the manuscript.

While it is very clear that the data supports the authors' claims, the authors also state that these genes are already transcribed in normoxia, what would happen in a gene that is on/off such as CA9 for example?

This is an interesting question. Existing literature (Xia, Genome Biology 2009; Choudhry, EMBO Rep 2014) has suggested the large majority of HIF target genes are already transcribed in normoxia, so we consider that our findings are likely to reflect the behaviour of the majority of the HIF transcriptome. However the referee is correct, CA9 is an unusual HIF target gene in respect of the on-off nature of its response and in MCF-7 cells demonstrates the highest-fold induction of any gene (Choudhry EMBO Rep 2014). Unfortunately in answering this question, the HIF binding site is right at the promoter and therefore it is not possible to examine the induction or otherwise of distant interactions between HIF and the promoter at this locus. However, we did examine looping from the NDRG1 promoter, which exhibits the 8th highest-fold induction of any gene in hypoxic MCF-7 cells (Figure S4) and observed distant interactions between the HIF binding site and the promoter in normoxic cells, which pre-dated HIF stabilisation.

The findings between HIF-1 and HIF-2 regulation of genes through their localisation at promoters or enhancer is very novel and interesting and should be placed in the main article, including Figure S6, and a list of the name of the genes analysed provided as supporting data.

We agree with the referee that differential regulation by HIF isoforms is interesting. However we note that referee 2 feels that this detracts from the main argument of the manuscript and suggested its omission. In our revision we have therefore left this figure in the supplementary data section, but we would be happy to be guided by an editorial perspective

We have added the requested data in Supplemental Table S2

Finally, just have a query as to the choice of 786-O cells for the analysis, since they only have HIF-2 and HIF-1. Sure, something like RCC4 would have been better to compare to MCF-7 that also has

both isoforms, was there a reason for this? It might have changed the results?

We chose 786-O cells because there are publically available datasets on chromatin accessibility, which are important to aid identification of functional elements on the chromatin. This type of data does not exist for RCC4 cells. Our data suggests that physical interactions between distant HIF binding sites and the target promoter(s) are independent of HIF-alpha isoform expression.

Referee #2:

In this paper, Platt et al. show that the majority of the epigenetic marks and chromatin loops surrounding hypoxia inducible genes are pre-established in normoxic conditions and that the activation of hypoxia-responsive transcription factor HIF does not cause dramatic changes in 3D genome organization. They conclude that HIF must be acting through already pre-existing genome structure to activate target genes, but that these existing loops can be used to determine which of numerous distal gene targets will be regulated by HIF in hypoxia. The approach, data, and analyses in this paper are mostly acceptable, and the question of how the genome organizes its response to this kind of signal is interesting, though numerous previous studies have looked at changes or lack of change in chromatin looping in other signal response systems in the past. Despite these previous studies, there is still some particular interest in examining changes in looping during hypoxia since chromatin structure changes have been implicated in this response in previous work. Also, though other papers have already shown the principle that quick signaling responses often do not involve changes in genome structure, this is still not accepted as "well known", so other evidence is useful.

All this being said, there are numerous points that require addressing to make sure that the authors are interpreting their data correctly and that their data support their conclusions.

The authors make some claims (detailed below) that are too strong for their data or which make too many unsubstantiated assumptions. My general concern is that the authors too freely assume that a high contact frequency in their data is the same thing as a specific "loop" (when some interactions are just part of a broader domain structure) and that a loop would mean direct regulation (when other evidence would be needed to show that a looping interaction causes regulation).

The referee correctly points out that the Capture-C methodology is detecting re-ligation of DpnII digested fragments resulting from a high contact frequency and that the presence of looping is indeed an inference rather than an observation. Although we have merely used terminology that is widespread in the field, we agree that it is imprecise and that these cis-interactions may result from a broader domain structure or compartmentalization. Furthermore, the term "loop" might falsely suggest that the interactions are always functional or that the intervening chromatin is not interacting at all. We have therefore revised the manuscript to more accurately reflect the observations made rather than any underlying assumptions.

The authors also need to do a better job of quantifying their comparisons (described in more detail below) rather than using either only visual inspection, anecdotal examples, or the strength correlation of called peaks. A more rigorous comparison is needed: can the authors compare how many peaks change between replicates vs. across conditions? If the changes are about the same, then indeed, nothing changes in hypoxia. Actual numbers of peaks that change in hypoxia or across cell types should be quantified. Further, in order to make the argument that these interaction patterns show "little change", it would be best to have an example of a case where similar profiles show a "substantial change". Can other literature be cited for this purpose? I am concerned that the differences in earlier work, where some papers show little change in interaction patterns and others do show changes, reflect more the pre-existing bias or subjective interpretation of different authors more than difference in data. So, the lack of difference claimed here needs to be carefully evaluated.

Thank you. All differences were in fact called on a statistical basis, based on two replicates. Nevertheless, we have performed extensive further quantitative analyses to address issues raised by the referee, both here and in response to later comments.

Firstly, as the referee points out in comment 27, the background Capture-C signal is influenced by distance-mediated decay. We had allowed for this when detecting sites of significant interaction, but

not when quantifying the strength of interaction at these sites. We have used this normalized data in all assessments of correlation between replicates or between conditions and provide this data in revised Figures 3, 4 and S1.

Secondly, as suggested in comment 14, we have extended our approach to identify and include all interactions and not just those that possess the assayed histone modifications. Although the biological significance of these sites is difficult to interpret, we agree that their inclusion provides a more complete analysis of the chromatin structure.

Though it is not possible to determine how many interacting regions have significantly different Capture-C signal between the two replicates of each condition (because the test of statistical significance requires there to be replicates), we have addressed the issue raised by the referee by plotting the correlation between the Capture-C signal for each replicate of each condition (now normalised to distance-decay, as requested in comment 27) in Supplemental Figure S1. This includes cis-interacting sites that interact in either or both dataset being compared, so that both quantitative and absolute differences in sites can be assessed. The correlation and distribution of values between each replicate is comparable to that between normoxia and hypoxia.

We accept the referee's point that further quantitation of the differences and comparison of situations in which differences were or were not observed could improve the clarity of our description. To address this we have provided quantitative data that enables comparison of differences between normoxia and hypoxia (MCF7 cells) with cell-type differences (between MCF-7 cells and 786-0 cells) in figures 4A and 4B of the revised manuscript. Only 3% of sites differ significantly (1.4-1.9 fold) between normoxia and hypoxia (highlighted in red in Figure 3B). Furthermore, the amplitude of these differences is small. In comparison, analysis of MCF-7 and 786-O Capture-C signal shows that approximately 20% of interacting sites differ significantly and that the amplitude of these differences is greater (1.2-6.0 fold) than those seen between normoxia and hypoxia (Figure 4 A and B).

The other major concern I have about the paper is the use of cancer cell lines. It is known that cancer cells vs. non-cancer cells respond differently to hypoxia, and that cancer cells often have to adapt to hypoxic tumor environments. So, it concerns me that the "pre-existing loops and histone marks" might be specific to the cancer cells studied. Can the authors provide evidence that primary cells also have pre-existing epigenetic marks poising them for hypoxic response?

The referee raises the suggestion that our observations in cancer cell lines might be specific to that setting and/or reflect adaptation to a hypoxic tumor environment. The possibility that primary or other cell types might behave differently and change with malignant transformation is certainly of interest, but distinguishing effects of cell of origin, transformed status and potentially long-term adaptation to hypoxia would require a very extensive piece of work outside the current study. Nevertheless the referee's point is interesting and we have added a caveat in discussion to highlight this point.

Also, just as a general point, in order to make this paper fit the mission of EMBO Reports (a single clear message), the authors should focus on supporting their main conclusion (that pre-existing looping defines potential target genes of HIF sites and the hypoxia response makes little change in this looping structure) and not get distracted by lots of other interesting data like whether some genes are regulated by either or both HIF1a and HIF1b. If these other points can be developed into something conclusive and interesting, then perhaps a different journal is appropriate, but in the current manuscript, they are a bit distracting (And, as I list below, some of the associated data is questionable).

We agree with the referee that differential regulation by HIF isoforms (we assume HIF-1alpha and HIF-2alpha is what is meant) is interesting. However we do think this data adds to the current manuscript in that this difference in activity of bound HIF isoforms is manifest despite the absence of changes in interaction. We also note that referee 1 has taken the opposite line to referee 2 and has asked for the data to be presented as a main figure. In our revision we have therefore left this figure in the supplementary data section, but we would be happy to be guided by an editorial perspective.

Specific Comments:

1) The Title (and claims elsewhere in the paper) make it sound like HIF only binds at locations where looped contacts to genes occur, while the data only show that many of the tested HIF sites have interactions with genes. This cannot be extrapolated to claim that all 1000 HIF sites in the genome loop to genes, or to prove that these interactions are regulatory. These claims should be reworded.

Yes we agree with the referee's perspective that the work does not imply (nor did we intend to imply) that all the thousands of HIF sites in the genome loop to genes and are functional. We have modified the title and the manuscript to avoid giving this impression (see also response comments below).

2) The statement in the introduction that "most intergenic HIF-binding regions do indeed function as transcriptional enhancers" is too strong. You say 9 of 14 with good signal interacted with at least 1 TSS (which were probably already pre-selected for being in good candidate regions). This does not cover the 100s or 1000s of sites genome wide. Nor does it show these actually function as enhancers.

We accept that the statement is too strong – at least if it were based solely on the evidence provided of physical interaction. We have observed robust statistical associations (i.e. across the pan-genomic range of HIF-binding sites) between promoter-distal HIF binding and hypoxic gene regulation (Schödel, Blood 2010). However we accept that this does not prove the function of each individual site. We have therefore amended the statement to read "Our findings revealed that these intergenic HIF-binding regions commonly interact with the promoters of hypoxia inducible genes".

3) The use of several qualifying phrases in the text seems unsupported. For example, on Page 3, how do you define a "surprisingly large" number of sites in an intergenic region? More than would be expected at random? Surprising only if we think no long distance regulation happens? More than for other TFs? Similarly, the statement that there are "very extensive cis-acting interactions" (page 4) is hard to justify: How do you define "very extensive"? More than at random regions? Or more than other non-hypoxic TFs during this response?

We accept this and agree that qualifying phrases many mean different things to different readers. Accordingly, we have revised the manuscript to simply give the figures without such additional phrases.

4) When the authors mention (Page 4) that hypoxia has been reported to affect genome structure, they should cite Kirmes et al., 2015 as a recent study on this topic.

We have now cited the relevant reference.

5) The authors state that Fig 1 shows that HIF binding occurs at enhancers that are already partially active in normoxic cells. But this statement comes from the average signal. Are there actually separable classes of enhancers, some of which are already active and some of which are not at all? What is the distribution around the averages shown in Fig 1? If enhancer regions are called in each condition, how many of them overlap between normoxia and hypoxia.

The referee is correct that average signal might hide distinct classes of site that behave differently and that this could be of interest. To address this we have plotted the frequency distribution for the \log_2 (fold-change in H3K27ac) and \log_2 (fold-change in RNAPol2) (see below). In each case, whilst the average signal is increased in hypoxia there is a considerable spread in the values, with the majority of sites increasing, but some not changing and some even going down. However, for both marks the distribution is unimodal around a positive log (fold-change), with no evidence to support two separate classes of sites (see below).

6) When the authors (page 6) speak of selecting HIF regions that "bore enhancer marks", are they speaking of normoxia enhancer marks? Hypoxia enhancer marks? A good negative control for their conclusions might be to do Capture C from one region that did not already have enhancer marks.

We apologize for not making the description of our methods clearer. The HIF regions themselves were chosen without reference to enhancer marks (DNA accessibility or histone modification), before the histone mark data was generated. They were subsequently found to bear these marks, which were present both in normoxia and in hypoxia (see Figure 1). We have amended the manuscript to remove this confusion.

The negative control data of the type the referee refers to was in fact reported in the Capture-C methods paper (Davies et al. Nature Methods 2015) where Capture C was performed from an inactive region of chromatin as suggested. This showed a symmetrical distribution of decreasing interaction frequencies indicative of only distance-dependent decay. Our statistical methods now take account of this distance-dependent decay, both when identifying sites of cis-interaction and when quantifying differences.

7) Page 7 (top): what are "functional domains"? Do the authors mean TADs? It would be more useful to evaluate how many interactions are within TADs, rather than within an arbitrary 500 kb.

We agree that the term "functional domains" is ambiguous and have removed it from the manuscript. We have performed the analysis suggested and find that all cis-interactions are within the same TAD as the bait site. We have included this in the revised manuscript.

8) Page 8 (top): "HIF can effect transcriptional regulation by looping..." This claim seems too strong: To prove this you would have to show that removing the particular enhancer would affect transcriptional regulation.

As previously, we accept that this statement is imprecise and have revised the manuscript to avoid the term 'looping' to state more precisely that promoter-distant HIF-binding sites can physically interact with hypoxia-regulated gene promoters.

9) Page 8 (middle): "suggests more extensive co-operativity between HIF and other transcriptional regulatory pathways" Maybe-but it could be alternately that all these other interactions are pre-set up

for totally independent pathways.

Yes, we agree that this is an alternative and not mutually exclusive possibility. We did not intend to imply that these must be working by modifying HIF itself or vice versa, although this may be the case in some instances. We have revised the manuscript to remove this ambiguity.

10) Page 11: The authors mention the "chromatin structure required for HIF activated gene expression", but they haven't directly shown that chromatin structure is required for HIF activated gene expression. It is possible that you could disrupt the structure and still see activation-- it is not proven otherwise here.

The referee is correct that whilst there is a lot of evidence that cis-interactions between enhancers and promoters are important for gene activation we have not formally tested the functional effects of disrupting chromatin structure at the loci whose structure we have analysed. This of course is not necessarily straightforward to perform or interpret. We have amended the manuscript to read "the chromatin structure between HIF-binding sites and HIF-activated promoters".

11) Page 12: it is never indicated which sites are bound by HIF in both cell lines. Should this be in a supplemental figure?

We have now included a Supplemental Table S1 listing all bait sites used in the Capture-C assays and indicating the cell type(s) in which they bound, the coordinates of the DpnII fragment captured, and the sequences of the oligonucleotides used to capture each site.

12) The finding about 3 genes looping to a co-regulated coding and non-coding gene pairing is quite interesting, but anecdotal and perhaps distracting from the main message of the paper

Although, as the reviewer points out, this could be regarded as anecdotal, the data is clear and we agree with the reviewer that it is interesting, hence its inclusion. We have deliberately kept this brief to avoid distracting from the main message.

13) A major question that is not addressed in this paper is: How do the interaction patterns of these HIF sites relate to other nearby non-HIF sites? Would any given captured region close by interact with all these marked regions? Or are these loops really specific to HIF sites? The authors could address this by trying a negative control capture of some non-HIF site within the general region of the genes and elements of interest. This would help address the concern that contacts might be general structural patterns rather than specific "loops" per-se.

There are potentially several questions in asking what would happen if we moved the bait region for the Capture-C away from the HIF binding site to nearby non-HIF binding sites. Clearly, bait regions that were very close to the HIF-binding site would be expected to give comparable patterns to capturing from the HIF site itself as the resolution of the technique is limited by the size of the DpnII fragments and the efficiency of cleavage, although in this respect Capture-C is the best resolution technique available. Within this limitation, Hughes et al (Nature Genetics, 2014) have captured from non-functional promoters within the alpha-globin gene cluster. Despite being within the same topologically associating domain as flanking functional promoters, these non-functional regions displayed distance-dependent background signals with no significant interaction with enhancers in the same general region. When Hughes et al, captured from different active regions within the alpha-globin TAD, comparable patterns of interaction were observed for each bait site, suggesting that all interacting sites within a TAD form a single interacting network. Finally, although not formally tested, we would expect a reciprocal nature to the interactions (i.e. if A interacts with B then B will interact with A). This is because, interactions are defined based upon re-ligation of A with B, so it would not matter which site was used to capture the re-ligated molecule. Indeed when we capture from HIF-binding sites we see interaction with HIF-regulated genes and when we capture from the promoters of HIF-regulated genes we see interaction with the HIF-binding sites. However, capturing from two mutually-interacting sites in the same experiment would not be informative in this regard, since it would be impossible to know which bait site had captured the re-ligated molecule.

Regarding the Methods section:

14) Generally, the analysis and normalization of the Capture-C data is thoughtful and solid. But, I am concerned about the use of "candidate interacting regions" based on epigenetic marks as a prior probability. It is a bit unclear in the methods, but it seems that they only considered candidates by epigenetic marks as possible regions of interaction. How can they then robustly draw conclusions about whether other looping interactions (that lack epigenetic marks) might change in hypoxia? The initial assumption that only marked regions can interact is not really proven in the literature. The authors should also do a comparison of all candidate interacting regions when they are comparing normoxia to hypoxia, and then focus on marked regions if appropriate.

Yes, this is a reasonable question. Consistent with previous studies (Hughes Nature Genetics 2014), the focus of our initial analysis was limited to promoters and regions bearing epigenetic marks, since we considered that the functional significance of these interactions would be easier to interpret. However, as requested we have now extended our analysis to include all significantly interacting sites, agnostic of chromatin marks. As anticipated by the referee, we have found additional interacting regions that were not marked by the histone modifications we assayed. Since our analysis of histone modifications is not exhaustive, it is difficult to know the functional or structural significance of these interactions (i.e. whether they might coincide with other 'chromatin marks' that were not assayed). However, importantly, we found these unmarked interactions also to be largely present before hypoxic induction of HIF. We have amended Figures 2B, 2C, 3B along with new figures 4A and 4B to include all interactions irrespective of the presence of marks.

15) How do the authors deal with a difference in sequencing coverage that may occur at certain locations in the genome due to a different frequency of restriction sites, different mappability, etc.?

The possible influence of the restriction enzyme fragment size on our analysis has several aspects. Firstly, the frequency of DpnII sites will affect the resolution of the technique. The smaller the fragments, the more accurately the cis-interacting region can be defined. Indeed, the higher frequency of DpnII (4-bp cutter) sites used in Capture-C compared to standard 6-bp restriction enzymes used in other approaches makes very large restriction fragments less common and the spatial resolution of Capture-C is generally superior to other techniques (Hughes, Nature Genetics 2014; Davies, Nature Methods 2016). Secondly, it might be anticipated that the number of reads mapping to a DpnII fragment would be proportional to its length and that this would therefore need to be factored in. However, Capture-C sonicates these fragments to an average size of 200 bp before sequencing and only DNA molecules that include paired-ends from separate DpnII fragments (one of which includes the bait region) are considered. Thus, sequences that are more than 200 bp from a DpnII site will be automatically excluded from analysis since their paired ends will be on the same DpnII fragment. Therefore the DpnII fragment size will have relatively little effect on the number of reads mapped to it.

We have examined the GC content of the terminal 200 bp of each DpnII fragment, both within statistically-significant cis-interacting regions and for all DpnII fragments that lie within 500 kb of a bait site. Although very similar, there was a slight excess of GC content in the cis-interacting regions than in background regions (mean 50% versus 48%). This is not surprising since potential cis-interacting regions are enriched for the presence of regulatory chromatin marks, which are enhanced within GC-rich regions. We therefore compared the GC-content of regulatory regions that interacted with bait sites with those that did not and have now included this in supplemental information (Figure S12A). This analysis shows that interacting and non-interacting regulatory regions had almost identical distributions of GC-content indicating that the identification of these regions is not substantially confounded by regions of differing GC-content.

Similarly, we have now checked the mappability of every DpnII fragment within 500 kb of each bait site by generating 50-base pseudo-reads every 9 bases within 200 bases of the end of each DpnII fragment in a manner similar to that previously described (Jin et al 2013). For each DpnII fragment we used bowtie to determine the fraction of uniquely mapped pseudo-reads expressed on a scale of 0 (no pseudo-reads mapped uniquely) to 1 (all pseudo-reads mapped uniquely). The proportion of DpnII fragments with varying degrees of mappability was then plotted as a frequency histogram for all DpnII fragments within 500 kb of a bait site and for DpnII fragments within statistically significant cis-interacting regions and we have now included this in supplemental information (Figure S12B). The majority of DpnII fragments revealed a high degree of mappability indicating

good coverage across the regions of interest. Furthermore, comparable levels of mappability were observed between the background regions and DpnII fragments within statistically significant cis-interacting regions, indicating that the identification of these regions is not substantially confounded by fragments of enhanced mappability.

16) Does the analysis approach for identifying pairs of reads eliminate undigested self-circles? (these will appear to be interactions > the 1 kb exclusion distance but might actually be the product of undigested self-circles?) Previous literature (ref 27 in this paper, Jin et al) shows that these artifacts can be prominent out to 5-10 kb. This may not be true for DpnII, but the authors should explicitly check for potential self-circle pairs.

This is an important possibility and we have therefore checked our data for the presence of undigested self-circles using the method outlined in Jin et al as suggested. As expected, this analysis shows an excess of outward orientated read pairs at short distances (gaps) between the pairs of reads consistent with the presence of small self-circles. However, for read pairs separated by 2 kb or greater the outward/same-strand and inward/same-strand ratios were both 0.5, consistent with efficient cutting and re-ligation and an absence of undigested self-circles of 2 kb or greater in size. This is a considerably shorter distance than was observed by Jin et al in their analysis of Hi-C data and likely arises from the higher frequency of DpnII sites compared to restriction enzymes that recognize 6-bp motifs. This will reduce the probability of a missed cleavage at a given distance from the bait site, since there will, on average, be more potential restriction sites per unit distance. Similar results were obtained for each replicate, condition and cell type and we have included these analyses as a new supplemental figure (Figure S10). Since all statistically significant cis-interactions were greater than 2 kb from the bait site we do not feel that the presence of small undigested self-circles of < 2 kb materially alters our findings.

17) "Normalized interaction frequency" is used in figures and referenced in the Methods without being carefully defined. Does this mean normalized by the distance decay? By the total number of reads in the dataset?

The interaction frequency is normalized to the total number of informative reads for a given bait region in any given experiment. This is defined as the total number of paired-end reads, for which one-end maps to the bait site DpnII fragment and the other end maps to an alternative DpnII fragment. In addition, we have now included normalization to the distance decay as suggested by the referee in comment 27. This has been clarified in the manuscript.

18) The manuscript indicates that CaptureC data are currently being submitted to GEO. This must be completed by publication date.

This has been completed. The Capture-C and new ChIP-seq data have been submitted to GEO under accession numbers GSE78100 and GSE78113 respectively and will be made public upon acceptance of the manuscript. The accession numbers have been added to the manuscript.

19) I do not see a record of what capture probes were used. This must be included.

This has been added as a supplemental table S1 (see comment 11).

20) A table of all determined interacting elements (FDR 5%) should be provided (with GEO or as supplement)

This has been added to the submission at GEO under GSE78100.

Figure 1:

21) Define better in the figure legend or methods where your datasets come from (all but ChIP-Seq pre-published, right?) What does "scaled" mean in the y-axis? Define a bit better somewhere.

The Capture-C (GSE78100), and ChIP-seq datasets for H3K4me1, H3K27ac and CTCF in MCF-7 cells and for H3K4me1, H3K4me3, H3K27ac, RNAPol2 and CTCF in 786-O cells (GSE78113) have not been previously published but have been deposited with the Gene Expression Omnibus at the

NCBI (see comment 18 above). ChIP-seq analyses of HIF-1alpha, HIF-2alpha and HIF-1beta ChIP-seq in MCF-7 cells (GSE28352), RNAPol2 and H3K4me3 ChIP-seq and RNA-seq in MCF-7 cells (E-MTAB-1994 and E-MTAB-1995), DNase-seq in MCF-7 cells (GSE32970), HIF-2alpha and HIF-1beta in 786-O cells (GSE67237), and FAIRE-seq in 786-O cells (GSM1011120) have been published previously and are all publically available as indicated. All datasets together with their accession numbers are now listed in the methods as requested.

“Scaled”, means scaled to the background signal for each dataset. We have defined this in the figure legend.

Figure 2:

22) It is unclear here and in a few other figures whether the Capture C shown here was done in hypoxia? Normoxia? Is the HIF binding from normoxia or hypoxia? (I assume hypoxia, else the paper interpretations make no sense, but it should be indicated).

The Capture-C and HIF-1alpha, HIF-2alpha and HIF-1beta ChIP-seq data originally shown in Figure 2 were performed in hypoxia only, whilst Figure 3 compared the normoxic and hypoxic Capture-C signal. In the light of this comment and those of referee 3, we have now included both normoxic and hypoxic datasets in Figure 2 and have annotated them clearly, to distinguish each.

23) 2D: label x-axis "gene rank for hypoxia induction" (or similar). What does color scale on x-axis mean? (ie are all pink/red induced and all blue repressed?) This is important to support the conclusion in the text that target promoters were "invariably" hypoxia-upregulated (that is only true if light pink is truly significantly upregulated)

The figure legend has been re-worded to provide greater clarity as suggested by the reviewer. The colour-scale is generated by the GSEA script to reflect the magnitude of the ranking metric, which is a combined measure of both the amplitude and statistical significance of the response (see Xiao Bioinformatics 2014 – ref 24). We have removed the word “invariably”, although all were statistically significantly upregulated, apart from one, which was just below the statistical threshold, when the amplitude of the change was factored out.

24) E: the authors say: "genes closest to HIF-binding sites, which did not loop to the site" Does this mean adjacent genes that did not loop? Or not necessarily adjacent but within a certain distance? In these cases, do these TSS lack active marks?

The referee is correct in their assumption that it is the adjacent genes that did not loop. We have amended the manuscript to remove any ambiguity. Whilst some of these TSS do lack active marks, others have ChIP-seq signals for H3K4me3, and RNAPol2 and produce transcripts by RNA-seq analysis indicating that they are active genes.

Figure 3:

25) I don't understand the point of Fig 3D. Why look at CTCF within TADs vs. everywhere? Conclusion from this is not explained.

We looked at CTCF binding frequency in normoxia and hypoxia focusing on the CTCF sites adjacent to the distal HIF binding sites studied (Fig 3C) and within the same TAD (Fig 3D). Changes in CTCF binding might suggest differences in local chromatin structure at these sites. However, we did not identify any changes in either situation (across two replicates). Upon consideration the data Fig 3D is partially redundant to that in Fig 3C, thus we have decided to remove Fig 3D.

Figure 4:

26) What are the highlighted genes? Known cell type specific ones? Is MYEOV not a cell type specific gene? Hypoxia induced? It is also cell type specific loop-why?

We have highlighted promoters to which the looping varies. The looping to MYEOV is not to the promoter of the gene, but is to an enhancer region just 3' to the gene. Given the large genomic distances from the HIF-binding sites it is difficult to represent this in a single figure. To clarify the issue we have now highlighted this site in a different colour and amended the figure legend.

Fig S1:

27) Reproducibility, here and elsewhere, is measured by interaction frequency comparison for "looping sites." The authors should check whether the relative levels of interaction among these sites are mostly dominated by distance-dependent decay (that is, the higher values are just closer to the capture site than lower values). If the dynamic range of these measurements is mostly distance dependent, then the fact that these interaction frequencies stay the same across conditions or replicates is not surprising. Check this by plotting correlation of interactions normalized by expected at distance, for example. Also, it would be very useful to report what percentage of called interactions in each replicate overlapped. This is true any time the authors compare two Capture C datasets: there needs to be a measure of how many peaks overlapped, not only how much shared peaks are similar in frequency.

The referee is correct in their assumption that the interaction frequency is a function (at least in part) of the distance from the bait site. Indeed, when determining interacting sites, we have used the average signal at a given distance from the bait site to estimate the background interaction frequency (see Figure S11). Although only sites at which the interaction frequency is statistically significantly higher than this background frequency have been called as interacting sites (see general comments above), this association with distance may still confound our analysis of reproducibility. We have therefore repeated our quantitative analyses (both of the signals from each replicate and when comparing different conditions) to include normalization to the background distance-decay as suggested and have revised supplemental Figure S1 and other figures accordingly (see above). These analyses confirm our previous findings showing a strong correlation between signals from each replicate.

Overall, we have chosen to represent the data quantitatively as interaction-frequency per site for each replicate or condition. These sites are defined in either or both datasets so that both unique and shared sites will be included in this analysis. This method was chosen because using algorithms to define sites in a binary (binding or not binding) fashion risks losing information and obscuring differences (where a site is called interacting in both datasets, but has a big quantitative difference) or falsely calling differences (where a site falls just above or just below a threshold in the two datasets, but where the difference is not statistically significant). Importantly, this analysis includes sites that were called significant in one condition only, but also takes into account any signal that was present in the "non-interacting" setting.

Fig S3:

28) What Chromosome is this? (state at top of figure) It appears that there is actually a decrease in peaks in hypoxia in 2 cases? (where HIF binds?) The authors need to indicate more carefully whether peaks were called significant in both normoxia and hypoxia. For this and other figures showing Capture C data, is the data presented the result of pooled replicates?

We apologize for the omission of the chromosome number. This has now been rectified.

Four HIF-binding sites were highlighted as significantly interacting with the NDRG1 promoter. Of these, only one showed a significant difference in normalized interaction frequency between normoxia and hypoxia.

For the reasons outlined above, we feel that reporting the number of peaks identified in one or other dataset or in both may be misleading (see response to general comments and comment 27) and prefer to use a more quantitative approach.

All statistical and quantitative analyses were performed on the average of the two replicates. However, the Capture-C tracks originally shown were taken from one replicate only. We have now re-plotted the tracks showing the average signal across the two replicates (see Figure S3).

29) Fig S5 C and D: need scale indicators-0 to 5 what? Kb? The authors are showing that distant enhancers are interacting even though HIF sites are proximal- this point would be even stronger if the authors checked if this is the same distribution of interacting enhancers distances that they saw for other promoter captures in this paper.

We have added a label to indicate that this is kb as requested.

Unfortunately, the number of other promoter captures is too small to permit a meaningful frequency distribution graph to be constructed. However, in all cases interactions well beyond 5 kb were detected (see Figure S3).

Fig S5 and S6:

30) Can the authors provide evidence that the HIF2a knockdown is working given that it usually has no effect in Fig S5?

The RNA-seq data for these analyses was taken from Choudhry et al EMBO Reports 2014. Both HIF-2alpha mRNA and protein levels were highly suppressed by HIF-2alpha siRNA in these experiments. In this earlier report, HIF-2alpha siRNA was observed to have a profound effect on hypoxic levels of specific non-coding RNAs. Furthermore, in the current report, HIF-2alpha siRNA has a suppressive effect on hypoxic levels of a number of mRNA transcripts at which the HIF-binding site is more than 10 kb from the promoter (Figure S6B).

31) All these knockdown heatmaps should be clustered to show different categories of single and combined effects. Right now, it is very hard to interpret the jumble of different responses.

We have performed hierarchical clustering of the rows as requested and have included the dendrograms in the revised Figures.

32) It is worrisome that some of the controls have such a variable Z score. (S6B in particular) I would assume that the control knockdowns should all have the same effect. How can the effect of a knockdown be known if the control is already affected?

The heat map analysis was performed using row normalization. Thus if any of the siRNAs leads to an increase in the transcript level, the control may be below the average for that row and therefore have a negative z-score. We have repeated the analysis normalised to the control siRNA condition for each gene and now represent the log₂ (fold-change) to better reflect the effects of the HIF siRNAs. This analysis is included in the revised Figures S5 and S6.

Discussion:

33) Can the authors speculate about how enhancers are pre-looped to the "correct" promoters (not the ones that aren't to be induced). There must be some other (HIF independent) mechanism of specifying poised promoters?

This is a very interesting question and one that is inherent to cell-type differentiation as well as the hypoxic response. It is likely that generic mechanisms, such as those involving CTCF and cohesin are important in maintaining the interactions, but what initially generates specific patterns of interaction is less clear. As the referee suggests, it likely involves HIF-independent cell-type or developmental transcription factors or other DNA-binding proteins that recognize specific DNA motifs. In this respect it is likely that analysis of chromatin interactions during cell differentiation would be most informative. As suggested we have added a comment in discussion to draw attention to this question.

34) The comments about 6,000 unique ligations, etc should move to the results section from the discussion.

This has been moved to earlier in the manuscript as requested.

35) It is a bit unfair to claim in the discussion that all but two HIF sites made physical contact with promoters when two of those were not actually called significant. Either the calling method has meaning and should be used consistently, or it doesn't and isn't useful.

We accept this and have revised the text to reflect just the number that were individually called significant.

36) If the genome architecture is all pre-defined, how can characterizing it reflect the "HIF

transcriptional response" (as claimed on page 14)? Doesn't it more accurately determine the landscape that the HIF response can work with?

The referee is correct in their statement. Indeed we feel that this is one of the main messages of the paper and have altered the language to that suggested by the referee.

Referee #3:

The manuscript addresses the mechanisms of transcriptional activation by HIF using Capture C to analyze interactions between enhancers and promoters under normoxia and hypoxia conditions. The conclusions offer important insights into the process by which paused genes release into productive elongation during activation. The following are some comments that should help the authors improve the manuscript:

1. On page 5, the authors describe that "ChIP-seq was performed in MCF-7 cells, incubated in normoxia or hypoxia (0.5% oxygen, 16 hr), using antibodies to H3K4me1 and H3K27ac and analyzed together with RNA-seq and H3K4me3, RNApol2 and HIF ChIP-seq datasets obtained under similar conditions ". This gives the impression that the distribution of HIF was also examined under normoxia conditions, but I could not find this information in the manuscript. For example, in Figure 1A, are the promoter-distal HIF-binding sites defined under normoxia or hypoxia conditions? I went back to previous manuscripts by the same authors where they published ChIP-seq data and this information is never given in any of these manuscripts. I am not an expert on HIF expression but I found several published papers indicating that HIF is transcribed under normoxia conditions, although the protein levels are regulated at posttranscriptional steps. This is a very important issue because it directly affects the main claim of the manuscript i.e. that HIF is recruited to preformed loops in hypoxia. However, if there is some HIF already present in cells under normoxia, then this claim may not be correct. The authors need to clarify this point and, in the process, write more descriptive figure legends.

We are grateful to the referee for drawing attention to this point. The referee is correct both in stating that HIF is regulated post-transcriptionally and that under some circumstances it may be functional in normoxic cells. However in these experimental conditions, protein levels of the oxygen-labile subunits, HIF-1alpha and HIF-2alpha, are essentially undetectable by Immunoblot in normoxic MCF-7 cells.

However, the referee also suggests (in comment 3) that even the very low levels of HIF-alpha present in normoxia might saturate HIF binding to chromatin and account for the constitutive looping that we observe. Inherently, this is unlikely, since HIF target genes are highly induced between normoxia and hypoxia. However, hypoxia also regulates the transactivating ability of HIF as well as its protein level, so we agree that it remains a formal possibility worthy of exploration. Therefore we have re-examined HIF DNA-binding in normoxia using ChIP-seq datasets for the HIF-1beta subunit, which is the common dimerization partner for both HIF-1alpha and HIF-2alpha isoforms (and detectable in ChIP-seq at the highest sensitivity) and have included this new analysis in Figure 1. This analysis revealed an almost complete absence of HIF-1beta binding signal at the promoter-distal HIF binding sites in normoxia when compared to the signal in hypoxia. Furthermore, restricted analysis of normoxic HIF-1beta binding at the specific promoter-distal bait sites used in the Capture-C experiments in this study revealed absence of normoxic signal at these sites (see revised Figures 2 and S2). These results substantiate our conclusion that cis-interactions involving HIF-binding sites are present in normoxia and do not require binding of HIF to these sites.

2. Page 5. "These promoter-distal HIF-binding sites were highly accessible compared to promoters, and were strongly enriched for the enhancer mark H3K4me1, but had low levels of the promoter mark, H3K4me3". I think it is well established that strong enhancers transcribe high levels of eRNAs and also contain H3K4me3.

The referee correctly points out that enhancers do frequently contain both H3K4me3 and H3K4me1 marks, enhancers have been characterized as having a higher ratio of H3K4me1 to H3K4me3 (ENCODE Consortium Nature 2012). We see evidence of H3K4me3 at these promoter-distal

enhancers although at a lower level than H3K4me1, we have altered the text to clarify this.

Whilst eRNAs are transcribed at high levels they are also highly unstable, thus making up a smaller proportion of reads in an RNA-seq experiment than their transcription rate would suggest. We were unable to confidently observe eRNAs. Most probably this was because our total RNA-seq data was not sequenced to sufficient depth. Alternative approaches studying actively transcribing RNA such as global run on sequencing (GRO-seq) would be more sensitive for detecting rapidly turning over eRNAs, but analysis of these transcripts was not the purpose of the study. To avoid confusion we have clarified this in the revised manuscript.

3. Page 5. "However, H3K27ac and RNAPol2 were also present at these sites in normoxia before HIF binds". As described under #1 above, it would be nice to show proof that this is true by showing ChIP-seq data with no HIF signal at these sites. Even if levels of HIF are low in normoxia as determined by Western analysis, it has been shown in various publications that when levels of a DNA binding protein are reduced by 90% using RNAi, the actual amount of protein bound to DNA does not change dramatically. I think this point has to be clearly addressed because it is the main conclusion of the manuscript.

Please see response to comment 1 and new data (Figure 1) as requested.

4. Page 6. "Capture-C allows an unbiased determination of distant interacting elements". Every experimental approach in biology is subject to biases.

Accepted and apologies. We merely wished to indicate that Capture-C can identify previously unknown sites of interaction in a non-hypothesis driven way in contradistinction to 3C, which requires testing of specific sites by qPCR. We have amended the manuscript to clarify this.

5. Figure 2A. The authors should discuss the fact that LUCAT1 does not appear to be expressed in spite of a strong interaction with the HIF site used as bait. In addition, the authors should show the structure of the ARRDC3 and ARRDC-AS1 in an expanded view so that each gene can be visualized separately.

Thank you. LUCAT1 is a lncRNA and consistent with other lncRNAs its expression is low compared to many mRNAs and hence difficult to see when displayed on the same scale as ARRDC3. However, LUCAT1 is expressed and does show hypoxic regulation. We have therefore amended Figure 2A to show both low-scale and high-scale RNA-seq tracks to illustrate the hypoxic induction of both transcripts. We have also expanded the schematic of the ARRDC3 and ARRDC3-AS1 loci to better demonstrate their overlap.

6. Page 6 and Figure 2A. The Capture C experiments displayed in this figure seem to suggest that there are additional interactions that appear quite strong based on the visual appearance of the data but were not called as significant by the computational approach used. I wonder if this could be due to the binning in 5 kb bins. Since the authors used DpnII to make the libraries, in principle they could use smaller bins if they had sufficient numbers of reads. I could not find information in the manuscript on how many QC'd reads were obtained for each experiment. This information should be presented in a supplemental table and discussed in the context of why interactions between HIF sites and promoters could not be detected in some cases (page 7).

The referee is correct that not all significantly interacting sites have been highlighted in this figure. Our original analysis was principally focused on interacting sites that coincided with functional elements as annotated by ChIP-seq analysis of H3K4me1, H3K4me3, H3K27ac, RNAPol2 and CTCF since their biological significance is easier to determine. As a consequence, only interacting sites bearing these marks were highlighted in Figure 2A and Figure S2&3. In the light of this comment and those of referee 2 we have repeated our analysis of the Capture-C data in a de-novo manner that is not based on the presence of histone modifications and have also reduced the binning to 2 kb as suggested. We now report cis-interacting sites for each bait region in an additional Supplemental Table S3 including the number of QC'd reads obtained for each bait site captured in each experiment as suggested. All sites are now included in our scatterplot analysis of differential cis-interactions in normoxia and hypoxia and between cell types (Figures 3 and 4).

7. Figure 3A. It is very difficult to appreciate the significance of the data presented in this figure. It appears that there are not significant changes in regions far from the bait. I wonder if it would be more informative to focus on the region surrounding the bait and try to examine interactions at high resolution i.e. either single fragment or 1 kb bins.

Thank you. The genomic width of the display window was selected to encompass the whole of the interacting domain for all of the sites. We agree that this and the distance-mediated decay in background signal makes the significance of more distant sites harder to appreciate in this panel. To minimize this we have reduced the width of the genomic regions shown in Figure 3A to expand the data as suggested. However, to do so further risks omitting these interactions from the view altogether. To help resolve these issues, we have revised our scatterplot analysis comparing normoxic and hypoxic signals to take account of the distance-mediated decay as suggested by referee 2. We have also revised our binning approach to use a smaller (2 kb) moving window as suggested by referee 3. However, it is not possible to reduce this window further, since the resolution is limited by the size of the DpnII fragments. In addition, we now highlight in red on the scatterplots those sites that show a statistically significant difference. Importantly, when both near and far interactions are included in this analysis, very few (approximately 3%) show a significant change between normoxia and hypoxia (Figure 3B). Furthermore, any changes seen are of small magnitude (1.4-1.9 fold) and of a quantitative rather than qualitative nature. By comparison Capture-C signals in MCF-7 and 786-O cells show many more significant differences (approximately 20%), which are also of greater magnitude (1.2-6.0 fold) (Figures 4A and 4B).

8. Page 10. : Neither CTCF binding close to HIF-binding sites, nor pan-genomic patterns of CTCF binding were significantly altered by hypoxia, suggesting that at least under these conditions, chromatin looping is not altered by induction of HIF (Figure 3C & D)". This is a very strong statement in the absence of any data to support it. Were the CTCF ChIP-seq experiments done in duplicate? How was the data analyzed to conclude that no alterations in CTCF binding were observed under hypoxia? It is possible that binding of CTCF is not affected but the distribution of cohesin is, and this regulates changes in looping.

The CTCF ChIP-seq experiments were performed in duplicate and the statement was based upon a statistical analysis of these results. However, the referee is quite correct to point out that constitutive CTCF binding is not the same as constitutive looping and that other factors also play a crucial role. We have amended the manuscript to reflect this.

9. Figure 4C. There seems to be at least one significant interaction in 786-O cells not present in MCF-7 cells that is not highlighted in the figure.

Yes this is correct. In figure 4, we had only highlighted cell-specific promoters that interact with the HIF-binding sites. The cis-interacting site referred to in Figure 4C at the MYEOV locus is actually just 3' to the gene at a site that has marks of an active enhancer, hence it was not highlighted. However we agree that this difference is of interest. We have clarified this in the figure legend and highlighted this site in a separate colour.

10. The conclusions of the manuscript would be more significant if the authors could address the issue of what is making the loops during normoxia. Cohesin and Mediator are two obvious candidates. Also, I feel that the authors do not emphasize sufficiently the fact that HIF-induced genes are paused, and that activation of these genes by recruitment of HIF to previously formed loops may be a strategy employed to release Pol II from paused genes, rather than a general mechanism for enhancer-induced transcription activation.

We agree that the question of what maintains the loops in normoxia is interesting. This is of course a major general issue in understanding chromatin function. Though interesting, we feel it outside the scope of the current manuscript. We also agree with the referee's point recruitment of HIF to previously formed loops at paused genes may be a strategy to release the paused Pol II. We have added a brief commentary on both these issues in the discussion.

I am very pleased to accept your manuscript for publication in the next available issue of EMBO reports. Thank you for your contribution to our journal.

REFEREE REPORTS

Referee #1

The authors have added significant explanation and changed some of the strong statements to reflect their findings. This is a very interesting piece of work which raises many more interesting questions. I have quite pleased with the revisions and I think the authors addressed the other reviewers comments quite reasonably as well. Given that EMBO reports does not need a finished story, I think this work is more than suitable for publication

Referee #2

In this revised manuscript, Platt et al. have clarified important ambiguous points in their data presentation and interpretation. The edits to the main figures are very helpful-- they now show normoxia and hypoxia for all genomic data, include all significant interactions rather than only those with relevant histone marks, and account for generic distance decay of their interaction profile. They have satisfactorily verified that the interaction changes they see between normoxia and hypoxia are indeed negligible compared to other conditions and have updated their figures and figure labels so that they are easier to interpret. They have also done a good job of modifying the language they use to describe their findings to make it more accurate. I also appreciate and agree with the authors' use of quantitative rather than binary measures of interaction, as emphasized in their point-by-point rebuttal. I am now satisfied that this papers findings are justified, well described, and important for the field.

The only question I had about the revised manuscript regards Figure 2. I noticed that while the distance distribution in 2C has changed in the revision vs. previous version (presumably due to updated interaction detection methods accounting for distance decay and including all regions rather than only certain histone modifications) and 2B changes accordingly, panel 2D is identical to its previous version. This likely indicates that while some interactions changed, the genes that interact with HIF binding sites stayed the same. But, I wanted to ask the authors to double check and make sure this panel did not need updating to match the revised interaction set. This would not require another round of revision, it is just something to double-check before publication.

Referee #3

The authors have addressed all the issues raised in the original reviews in a very thoughtful and convincing manner. I believe this represents a nice piece of work and is appropriate for publication in EMBO Reports.

Corresponding Author Name: DAVID MOLE
Journal Submitted to: EMBO REPORTS
Manuscript Number: EMBOR-2016-42198V2